# Diet-driven differential response of *Akkermansia muciniphila* modulates pathogen susceptibility

Mathis Wolter[1,2], Erica T Grant [ID] [1,2], Marie Boudaud [ID] [1], Nicholas A Pudlo[3], Gabriel V Pereira[3], Kathryn A Eaton[3], Eric C Martens[3] & Mahesh S Desai [ID] [1,4] [✉]

## Abstract

The erosion of the colonic mucus layer by a dietary fiber-deprived gut microbiota results in heightened susceptibility to an attaching and effacing pathogen, *Citrobacter rodentium*. Nevertheless, the questions of whether and how specific mucolytic bacteria aid in the increased pathogen susceptibility remain unexplored. Here, we leverage a functionally characterized, 14-member synthetic human microbiota in gnotobiotic mice to deduce which bacteria and functions are responsible for the pathogen susceptibility. Using strain dropouts of mucolytic bacteria from the community, we show that *Akkermansia muciniphila* renders the host more vulnerable to the mucosal pathogen during fiber deprivation. However, the presence of *A. muciniphila* reduces pathogen load on a fiber-sufficient diet, highlighting the context-dependent beneficial effects of this mucin specialist. The enhanced pathogen susceptibility is not owing to altered host immune or pathogen responses, but is driven by a combination of increased mucus penetrability and altered activities of *A. muciniphila* and other community members. Our study provides novel insights into the mechanisms of how discrete functional responses of the same mucolytic bacterium either resist or enhance enteric pathogen susceptibility.

**Keywords** *Akkermansia muciniphila*; Dietary Fiber; Synthetic Gut Microbiota; Community Ecology; *Citrobacter rodentium*
**Subject Category** Microbiology, Virology & Host Pathogen Interaction

## Introduction

The intestinal mucus layer is a protective and lubricating barrier of mucin glycoproteins covering the intestinal epithelium that is an integral part of the mucosal immune system (Johansson and Hansson, 2016). Among the many functions of the gut mucus layer is the protection against enteric pathogens (Martens et al, 2018). The colonic mucus layer is mainly composed of mucin-2 (MUC2) glycoproteins, which are dynamically secreted by goblet cells in a

greatly condensed form to generate a nearly impenetrable, net-like structure covering the epithelium (Johansson et al, 2008). Some members of the gut microbiota possess enzymatic capabilities to degrade the complex mucin glycoproteins (Parrish et al, 2022a; Martens et al, 2018) and constantly feed on the outer edges of this structure, resulting in a progressively looser layer further in the lumen (Johansson et al, 2008). Thus, the mucus layer is an important nutrient source for the gut microbiota (Schroeder, 2019; Luis and Hansson, 2023) and a critical interface between the microbiota and the immune system (Paone and Cani, 2020; Wolter et al, 2021a).

The observation that a dietary fiber-deprived gut microbiota deteriorates the colonic mucus barrier has been replicated across labs, among mice harboring either complex or simplified microbial communities (Desai et al, 2016; Schroeder et al, 2018; Riva et al, 2019; Neumann et al, 2021; Parrish et al, 2023; Kuffa et al, 2023; Pereira et al, 2024). Moreover, we have shown previously that a fiber-deprived gut microbiota enhances susceptibility to an attaching and effacing, rodent mucosal pathogen, *Citrobacter rodentium* (Neumann et al, 2021; Desai et al, 2016), which is an important model pathogen for human enteropathogenic and enterohemorrhagic *Escherichia coli* (Mullineaux-Sanders et al, 2019). It is striking that a typically self-limiting pathogen such as *C. rodentium* leads to lethal colitis in fiber-deprived mice with a microbiota (Neumann et al, 2021; Desai et al, 2016). This lethal colitis was not observed in fiber-deprived, germ-free mice that also exhibit a thinner colonic mucus barrier and a similar invasion of the pathogen in the mucosa as susceptible mice (Desai et al, 2016). Thus, it is evident that the pathogen alone is incapable of driving the lethal phenotype and an interaction of the microbiota with the pathogen plays a critical role.

Nevertheless, the mechanisms connecting the excessive mucus degradation, increased pathogen susceptibility, and specific microbial members remain unknown. To elucidate which specific bacteria or functions are responsible for increasing pathogen susceptibility in the absence of fiber, here we leverage a community ecology approach by performing multiple strain dropout experiments using a functionally characterized 14-member synthetic human gut microbiota in gnotobiotic mice. We identify a commensal microbe, *Akkermansia muciniphila*, to play a vital role in modulating infection dynamics of *C. rodentium* in a

[1]Department of Infection and Immunity, Luxembourg Institute of Health, Esch-sur-Alzette, Luxembourg. [2]Faculty of Science, Technology and Medicine, University of Luxembourg, Esch-sur-Alzette, Luxembourg. [3]Department of Microbiology and Immunology, University of Michigan Medical School, Ann Arbor, MI, USA. [4]Odense Research Center for Anaphylaxis, Department of Dermatology and Allergy Center, Odense University Hospital, University of Southern Denmark, Odense, Denmark. [✉]E-mail: mahesh.desai@lih.lu

diet-dependent manner. Our study provides valuable lessons to better understand the pathogenesis mechanisms of an attaching and effacing pathogen by incorporating a community ecology perspective.

# Results

## Strain dropout approach provides functionally relevant communities to study enteric infection

We colonized age-matched, germ-free Swiss Webster mice with different synthetic microbiota (SM) communities and maintained these mice on a standard lab chow, which we label a fiber-rich (FR) diet (Fig. 1A). Based on our previous work (Desai et al, 2016), we sought to tease apart how and which mucin-degrading bacteria are directly responsible for the increased susceptibility to *C. rodentium* during dietary fiber deprivation. We designed four new SMs by dropping out different mucolytic bacteria from the 14SM community (Desai et al, 2016) (Fig. 1B). The 10SM contains none of four mucolytic bacteria present in the 14SM: *Akkermansia muciniphila*, *Barnesiella intestinihominis*, *Bacteroides caccae*, and *Bacteroides thetaiotaomicron* (Fig. 1B). The 11SM contains all members of the 10SM plus *A. muciniphila*, a mucin-specialist bacterium that can utilize mucin *O*-glycans as the sole carbon and nitrogen source (Derrien et al, 2004; Desai et al, 2016). The 12SM consists of 10SM plus *B. caccae* and *B. thetaiotaomicron*, both of which are mucin generalists, meaning they can grow on a number of polysaccharides, including dietary fibers and mucin *O*-glycans (Desai et al, 2016) (Fig. 1B). Finally, the 13SM contains both mucin generalists in addition to the mucin-specialist *B. intestinihominis*; therefore, only *A. muciniphila* is excluded from the community.

Initial colonization of each SM community was verified by qPCR using primers specific to each of the 14 bacterial strains as previously described (Desai et al, 2016; Steimle et al, 2021). Mice were maintained for 14 days on the FR diet before approximately half of the mice in each experimental group were switched to a fiber-free (FF) diet (Fig. 1A); the mice were maintained on the respective FR and FF diets for forty days. Forty days after the diet switch, a subset of mice from each group was sacrificed to collect the pre-infection readouts (Fig. 1A). The relative abundances of different members of the SM were determined throughout the experimental timeline by 16S rRNA gene sequencing (Fig. EV1), with the averages for each group prior to the infection depicted in Fig. 1C. Across different SM combinations, two consistent shifts could be observed in the FF group, relative to the FR group: (1) an expansion of the mucolytic bacteria *A. muciniphila* and/or *B. caccae* (when present in the SM); and (2) a reduction of fiber-degrading bacteria such as *Eubacterium rectale* and *Bacteroides ovatus* (Figs. 1C and EV1). These microbial changes resulted in an overall increased relative abundance of mucolytic bacteria in place of the non-mucin-degrading bacteria during the FF diet (Fig. 1D), with the majority of the non-mucin-degrading population being fiber degraders.

Even without the mucolytic bacteria, the SM form a fairly stable community (Fig. EV1; Dataset EV1), which is further emphasized by examining the fold-changes of the individual strains between diets just prior to infection (Fig. 1E). We observed similar changes

between diets across different SM combinations. Based on the microbial abundance data, it seems that of the two mucin-degrading bacteria, *A. muciniphila* and *B. caccae* are favored, or at least not adversely impacted, under FF conditions (Fig. 1E). In contrast, *B. thetaiotaomicron* and *B. intestinihominis* seem to be outcompeted by the other mucin-degrading bacteria under FF conditions, resulting in a relatively lower abundance under these conditions (Fig. 1E). Among the non-mucin-degrading bacteria, *Desulfovibrio piger*, *Clostridium symbiosum* and *Collinsella aerofaciens* also thrived under FF conditions (Fig. 1E), which, in the case of *C. symbiosum* and *C. aerofaciens*, may be a result of either free amino acids and other liberated metabolites that are not utilized by the primary mucin degraders (Fischbach and Sonnenburg, 2011) or in case of *D. piger* may be owing to the release of terminal sulfate groups of mucins (Rey et al, 2013). Of these, *C. aerofaciens* has been associated with a number of inflammatory diseases, in particular, rheumatoid arthritis (Chen et al, 2016a). It is also known to be associated with *D. piger* as it produces lactate, $H_2$, and formate, which support growth of *D. piger* (Rey et al, 2013) and contribute to luminal $H_2S$, which can damage the mucosa at high concentrations (Blachier et al, 2021).

In order to assess the presence of any baseline inflammation, which may be induced by the diet shift, we measured pre-infection fecal lipocalin-2 (LCN-2) concentration, which is a marker of low-grade inflammation (Fig. 1F) (Chassaing et al, 2012). The 14SM—and, to a lesser extent, the 11SM and 10SM groups—showed statistically meaningful increases in LCN-2 on the FF diet (Fig. 1F). Meanwhile, only the 14SM, 13SM, and 12SM groups showed a significant shortening of the colon on the FF diet, which is another gross indicator of inflammation (Fig. 1G). It is worth noting that the 10SM mice exhibited shorter colons under both diets, underscoring that mucin-degrading microbes can reduce inflammation and promote epithelial barrier integrity in the context of a fiber-sufficient diet; our in vivo results provide evidence for the earlier data from in vitro models that mucin-degrading bacteria improve markers of barrier integrity (Pan et al, 2022). While direct observation of all mice showed no overt signs of disease, these results suggest that dietary fiber deprivation has the potential to induce low-grade inflammation, which appears to be further dependent on the community composition.

Considering the vital role of short-chain fatty acids (SCFAs) in promoting intestinal homeostasis (van der Hee and Wells, 2021) and susceptibility to *C. rodentium* (Osbelt et al, 2020; An et al, 2021), we suspected that possible shifts in the microbial SCFA production might alter responses to *C. rodentium* after infection. Although no differences were observed for acetate between the two diets in all SM groups, butyrate concentrations were significantly different for all dietary groups except 10SM (Fig. 1H). Interestingly, among the two groups containing *A. muciniphila* (14SM and 11SM), concentrations of cecal propionate—a metabolite produced by *A. muciniphila* (Derrien et al, 2004)—were reduced in FF groups (Fig. 1H). This observation is in contrast to the results of the complete community lacking *A. muciniphila* (13SM). These results suggest that the metabolism of this mucin specialist is altered in FF mice compared to FR mice. Importantly, our strain dropout approach successfully provides functionally relevant stable communities to possibly identify causal mucin-degrading microbes that might aid the increased susceptibility to *C. rodentium* in fiber-deprived mice.

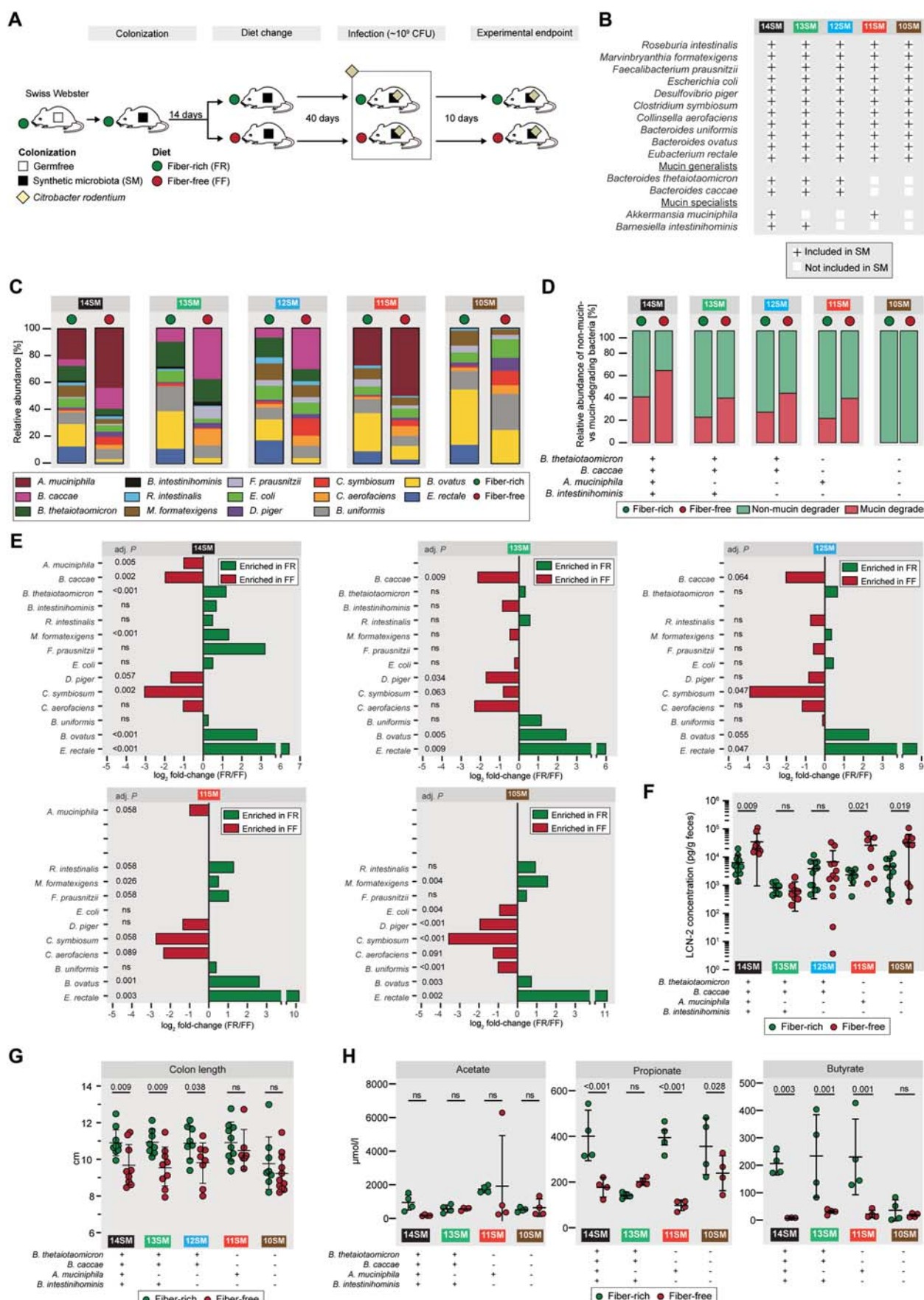

**Figure 1. Species-dropout approach enables distinct functionally characterized synthetic communities in vivo.**

(A) Experimental timeline. Age-matched germ-free Swiss Webster mice were gavaged with one of the different synthetic gut microbiota (SM) on three consecutive days. These mice were maintained for 14 days on the fiber-rich (FR) diet, after which approximately half of the mice were switched to a fiber-free (FF) diet. The mice were maintained for 40 days on their respective diets; all readouts in this figure were from samples on day 40, before they were infected with *Citrobacter rodentium*. After infection, mice were closely observed for up to 10 days. For uninfected readouts: one (13SM, 12SM, and 11SM) to two independent experiments (14SM and 10SM) were performed for both diets; for infected readouts, an additional one (13SM and 11SM) to two independent experiments (14SM, 12SM, and 10SM) were performed for both diets. (B) Table representing five different synthetic microbiota compositions (10SM, 11SM, 12SM, 13SM, and 14SM). Bacteria which are present in a given SM are represented by a plus (+) sign. (C) Relative abundance of the gut bacteria as determined by 16S rRNA gene sequencing of stools of uninfected mice. (D) The relative proportion of mucin- and non-mucin-degrading bacteria in each of the groups. (E) FR/FF fold-changes of the relative abundance of the gut bacteria as determined by 16S rRNA gene sequencing of stools of uninfected mice after 40 days of feeding a FR or FF diet ($n = 3$–9 per group following removal of four outliers each from 14SM FR and 14SM FF, 2 from 13SM FF, and 1 from 11SM FF). Relative abundance values were analyzed by multiple unpaired *t*-tests with Welch correction and with *p* value adjustment using the Benjamini–Hochberg method. (F) Fecal lipocalin-2 (LCN-2) concentration (pg/g feces) of uninfected mice ($n = 7$–12 per group, no outliers). Log-transformed concentrations were analyzed by two-way ANOVA with adjustment for multiple comparisons between SMs of the same diet (shown) and between diets of the same SM (see Source Data) using the Benjamini–Hochberg method. Error bars show SD and the center is the arithmetic mean. (G) Colon lengths of uninfected mice ($n = 7$–10 per group, no outliers), analyzed by two-way ANOVA with adjustment for multiple comparisons between SMs of the same diet (shown) and between diets of the same SM (see Source Data) using the Benjamini–Hochberg method. Error bars show SD and the center is the arithmetic mean. (H) Cecal short-chain fatty acid (SCFA) concentrations of uninfected mice ($n = 4$ per group, one outlier removed from butyrate 14SM FF). Median concentrations from triplicate measurements were analyzed by two-way ANOVA with adjustment for multiple comparisons between SMs of the same diet (shown) and between diets of the same SM (see Source Data) using the Benjamini–Hochberg method. Error bars show SD and the center is the arithmetic mean. ns non-significant (adjusted *p* > 0.1). Source data are available online for this figure.

## *A. muciniphila* is responsible for increased *C. rodentium* susceptibility during fiber deprivation

The remaining mice (after those sacrificed for pre-infection readouts) were infected with ~$10^9$ CFU of *C. rodentium* (Fig. 1A). The *C. rodentium*-infected mice were monitored for up to 10 days post-infection (DPI) before they were sacrificed to collect final readouts (Fig. 1A). Overall, we could reproduce the higher pathogen susceptibility phenotype previously described in the 14SM colonized mice after 40 days of feeding on an FF diet (Desai et al, 2016). Comparing *C. rodentium* loads between colonization states, within the same diet, we reached an interesting conclusion that *A. muciniphila* tends to confer resistance to infection, but only under fiber-rich conditions and in the absence of other mucin degraders in the community, that is, in the 11SM condition (Fig. 2A,B). Yet when fed an FF diet, the 11SM-colonized mice showed increased *C. rodentium* levels, as observed in the 14SM group (Fig. 2C,D). The 13SM, 12SM, and 10SM groups, which lacked *A. muciniphila*, also showed elevated pathogen loads on the FF diet according to AUC (Fig. 2C), potentially reflecting a generalized impact of diet on the susceptibility; however, this difference was non-significant in most cases for daily comparisons (Fig. 2D).

The maximum weight loss following infection and fecal LCN-2 at the end of the infection were mostly in alignment with the pathogen loads, with the 14SM and 11SM groups showing significant weight loss and increased LCN-2 levels on FF diet (Fig. 2E,F). After infection, the 10SM-colonized mice exhibited higher levels of LCN-2 during fiber deprivation (Fig. 2F). As this is in line with their levels of LCN-2 prior to infection (Fig. 1F) and their diet-dependent differences in pathogen load (Fig. 2C,D), these data suggest that a complete absence of mucin-degrading bacteria is also a precursor to increased pathogen-induced inflammation in the context of a fiber-deficient diet. The colon lengths after the infection were reflective of the pre-infection status, with the 14SM, 13SM, and 12SM groups showing a significant shortening on the FF diet (Fig. 2G). Histological analysis of the cecal tissue corroborated the maximum weight loss during fiber deprivation in the 14SM and 11SM groups; however, we also observed elevated disease scores by histology in all groups on the FF diet, relative to the FR diet

(Fig. 2H). Overall, our data indicate that, during fiber deprivation, the increased susceptibility to *C. rodentium* in the presence of mucolytic bacteria is exacerbated by a single mucin specialist, *A. muciniphila*. Furthermore, the presence of all other mucin-degrading bacteria seems to amplify the physiological impact of *A. muciniphila*, while the lack of *A. muciniphila* is sufficient to avoid the excessive vulnerability to *C. rodentium* observed on the FF diet.

As *A. muciniphila* abundance is indirectly modulated by the presence of dietary fiber, we supplemented the FF diet with 7.5% acetylated galactoglucomannan (AcGGM), a purified plant polysaccharide (Fig. EV2A). Although AcGGM is primarily degraded by *Roseburia intestinalis* (La Rosa et al, 2019b), a butyrogenic commensal present in the SM communities, we did not detect an increase in the relative abundances of these bacteria, but rather saw an expansion of *Bacteroides uniformis* (Fig. EV2A), which can be explained by its demonstrated capacity to subsist on this substrate (La Rosa et al, 2019a). Although we aimed to rescue the vulnerable 14SM phenotype by supporting the growth and metabolic activity of the aforementioned butyrogenic commensals, the AcGGM supplemented group showed similar *C. rodentium* loads (Fig. EV2B), LCN-2 levels (Fig. EV2C) and colon lengths (Fig. EV2D) as the 14SM FF group. In line with our earlier study suggesting that purified fibers do not alleviate the excessive microbial mucin foraging (Desai et al, 2016), the current results corroborate the conclusion that purified prebiotic supplementation is insufficient to prevent the higher pathogen susceptibility observed among FF-fed mice. This experiment serves as an important cautionary tale: the functional activities as predicted from single strains grown in monoculture do not necessarily translate into the same overall effect in a more complex community, therefore it is crucial to test prebiotic supplements using complex communities or defined consortia *in vivo* in order to validate their intended effect.

We then sought to determine whether the FF diet led to increased *C. rodentium* load in 14SM mice could also be seen with shortened feeding timelines, as we expected that the time spent on the FF diet might influence the detrimental impact on the host. Thus, we tested whether 5 or 20 days on a fiber-deprived diet could elicit a similar pathogen load. Overall, *C. rodentium* loads of the shorter FF feeding groups (5 and 20 days) were comparable to a

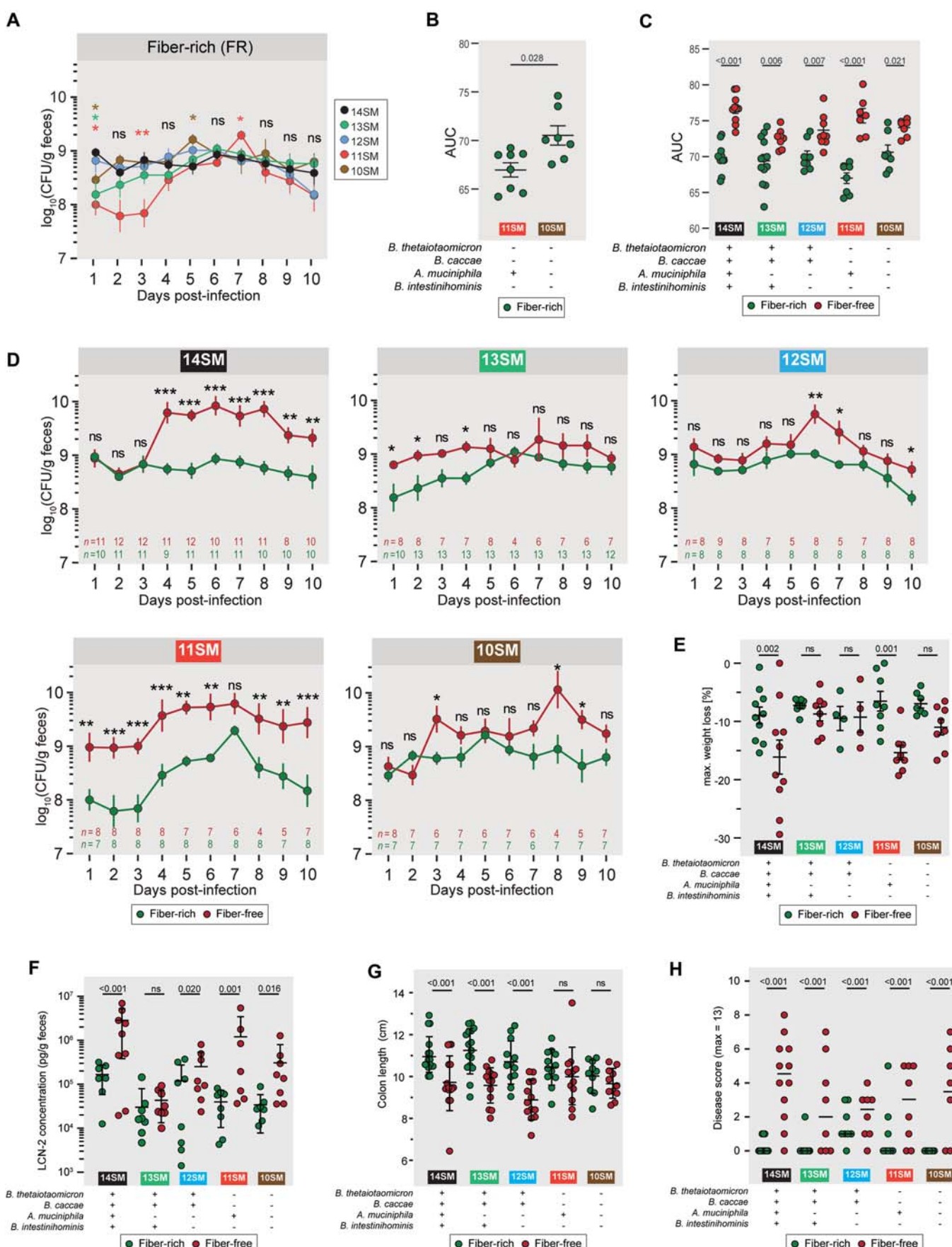

**Figure 2.  A. muciniphila drives increased C. rodentium susceptibility during fiber deprivation.**

(A) Fecal *C. rodentium* load of gnotobiotic Swiss Webster mice from 1–10 days post-infection (DPI) among mice fed a standard, fiber-rich (FR) diet to compare between SM compositions. One (13SM and 11SM) to two independent experiments (14SM, 12SM, and 10SM) were performed for both diets. Log-transformed CFU/g were analyzed using a mixed effects model with matching on mouse ID, Geiger–Greenhouse correction, and *p* value adjustment using the Benjamini–Hochberg method with 14SM as the control for multiple comparisons. The number of datapoints are indicated in panel (D); six outliers were excluded. Error bars show SEM and the curves the arithmetic mean. ns non-significant; *, adjusted *p* < 0.1; **, adjusted *p* < 0.01 (see Source Data for exact values). The asterisk color represents the group exhibiting a significant difference compared to 14SM. (B, C) Area under the curve (AUC) for fecal *C. rodentium* loads across the 10 days post-infection for (B) 11SM and 10SM FR mice and (C) each SM and diet combination. Error bars show SD and the center the arithmetic mean. Two-way ANOVA with adjustment for multiple comparisons using the Benjamini–Hochberg method between (B) SM or (C) diets. Missing values were imputed by calculating the geometric mean of adjacent time points or assuming the value of the adjacent timepoint if the missing value occurred at the start of the series. In cases where ≥2 values were missing from the end of the sampling period, the mouse was excluded from the AUC analysis. (*n* = 6–13 per group, following exclusion of outliers: 14SM FF, 2 excluded; 11SM FF, 1 excluded; 10SM FF, 1 excluded). (D) Fecal *C. rodentium* load of gnotobiotic Swiss Webster mice from 1–10 days post-infection (DPI) split into plots according to SM colonization to compare between diets. Log-transformed CFU/g were analyzed using a mixed effects model with matching on mouse ID, Geiger–Greenhouse correction, and *p* value adjustment using the Benjamini–Hochberg method. The number of datapoints are indicated on each graph; six outliers were excluded. Error bars show SEM and the curves the arithmetic mean. ns, non-significant; *, adjusted *p* < 0.1; **, adjusted *p* < 0.01; *** adjusted *p* < 0.001 (see Source Data for exact values). (E) Maximum weight loss during *C. rodentium* infection (*n* = 4–10 per group, no outliers), analyzed by two-way ANOVA with adjustment for multiple comparisons between SMs of same diet (shown) and between diets of same SM (see Source Data) using the Benjamini–Hochberg method. Error bars represent SD and the center is the arithmetic mean. (F) Fecal LCN-2 concentration (pg/g feces) of infected mice at 10 DPI (*n* = 7–8 per group, no outliers). Log-transformed concentrations were analyzed by two-way ANOVA with adjustment for multiple comparisons between SMs of same diet (shown) and between diets of same SM (see Source Data) using the Benjamini–Hochberg method. Error bars show SD and the center is the arithmetic mean. (G) Colon length of infected mice at 10 DPI (*n* = 10–16 per group, no outliers), analyzed by two-way ANOVA with adjustment for multiple comparisons between SMs of same diet (shown) and between diets of same SM (see Source Data) using the Benjamini–Hochberg method. Error bars show SD and the center is the arithmetic mean. (H) Histologically determined disease score from cecal tissue of infected mice 10 DPI (*n* = 8–12 per group), analyzed by two-way ordinal regression with *p* value adjustment using the Benjamini–Hochberg method. Bars represent the median. Source data are available online for this figure.

40-day FF diet feeding period (Fig. EV2E–G). Minor differences in pathogen loads are most likely attributable to the low sampling number of the 5- and 20-day groups (*n* = 4 and *n* = 3, respectively). These results suggest that the local microbiota effects, including mucus–microbe interactions, rather than the detrimental impact on the host owing to longer feeding periods (like 40 days), might play decisive roles in the increased pathogen susceptibility.

## Host immune and pathogen transcript readouts suggest a differential response of *A. muciniphila* behind increased susceptibility

As *A. muciniphila* has been studied for its potential immunomodulatory properties (Luo et al, 2021; Ansaldo et al, 2019; Ashrafian et al, 2021; Xie et al, 2023; Zhang et al, 2023; Grant et al, 2023; Parrish et al, 2023), we supposed that the relative expansion of *A. muciniphila* on the FF diet might have repercussions on the host immune response, thus explaining the increased *C. rodentium* susceptibility. As the initial expansion of *C. rodentium* is heavily affected by dietary-fiber deprivation, we chose to assess the immune cell populations at 3 DPI using fluorescence-activated cell sorting (FACS). We have previously observed that by 4 DPI, the pathogen successfully enters the intestinal tissue (Desai et al, 2016); therefore, we surmised that 3 DPI would allow us to capture the early innate immune responses known to be driven by *C. rodentium* (Mullineaux-Sanders et al, 2019) before the complete invasion of the pathogen. We also used the same time point (3 DPI) to study both host colonic- and microbial transcriptomes. In order to facilitate comparison with our previous work (Desai et al, 2016), we continued using a 40-day FF feeding period for this and subsequent experiments in the current study.

Overall, different immune cell populations showed similar trends under fiber deprivation, but nearly all comparisons were statistically non-significant after correction for multiple comparisons (Fig. EV3A,B). In particular, the RORγt-positive T helper cell population (CD4$^+$RORγt$^+$) was significantly decreased in the FF

diet groups compared to the FR groups (Fig. 3A). As Th17 cells play an essential role in the clearance of *C. rodentium* (Atarashi et al, 2015), these data point to a lack of Th17 induction as a possible contributor to the increased pathogen susceptibility observed among the 14SM FF diet group compared to the 14SM FR. However, comparing 14SM and 13SM under the same diet revealed no statistically significant shifts in proportions of the immune cell populations assessed here, prompting us to investigate other facets of the host–microbiome axis for explanations of the altered susceptibility.

To further investigate the role of *A. muciniphila* in modulating pathogen susceptibility, we analyzed both mouse colonic tissue (3 DPI) and *C. rodentium* transcripts (3 DPI, from microbial metatranscriptomics data) in both diet groups of the 13SM and 14SM mice. As with the FACS data, there was considerable variability in the host transcriptomic data, which is why no distinct clustering by group could be observed in the Principal Components Analysis (PCA) plot (Fig. 3B). In contrast, *C. rodentium* transcripts showed clear clustering based on diet (Fig. 3C). Analysis of the host transcript counts (Dataset EV2) showed only a limited number of genes differentially transcribed between the four groups (Dataset EV3). Within the 13SM group, 19 host genes were upregulated in the FR group and 8 in the FF group (Fig. 3D; Dataset EV3). Meanwhile, in the 14SM group, 9 host genes were upregulated during the FR diet and 13 during the FF diet (Fig. 3D; Dataset EV3). Only two transcripts, C4b—a complement factor involved in innate immune response elevated in FF-fed mice—and Klf6—a key regulator of pathogenic myeloid cell activation increased in FR-fed mice—were significantly altered in both the 13 and 14SM groups (Dataset EV3). Evaluating the effects of the presence of *A. muciniphila* in the same fiber-deprived conditions, revealed enrichment of transcripts involved in immunoglobulin production (Igkv12–89, Igkv1-110) and zinc homeostasis (Slc39a4) in the 14SM FF group, which are both associated with infection response; however, it is unclear how increased transcription of these genes might lead to increased susceptibility to infection. Overall, the host

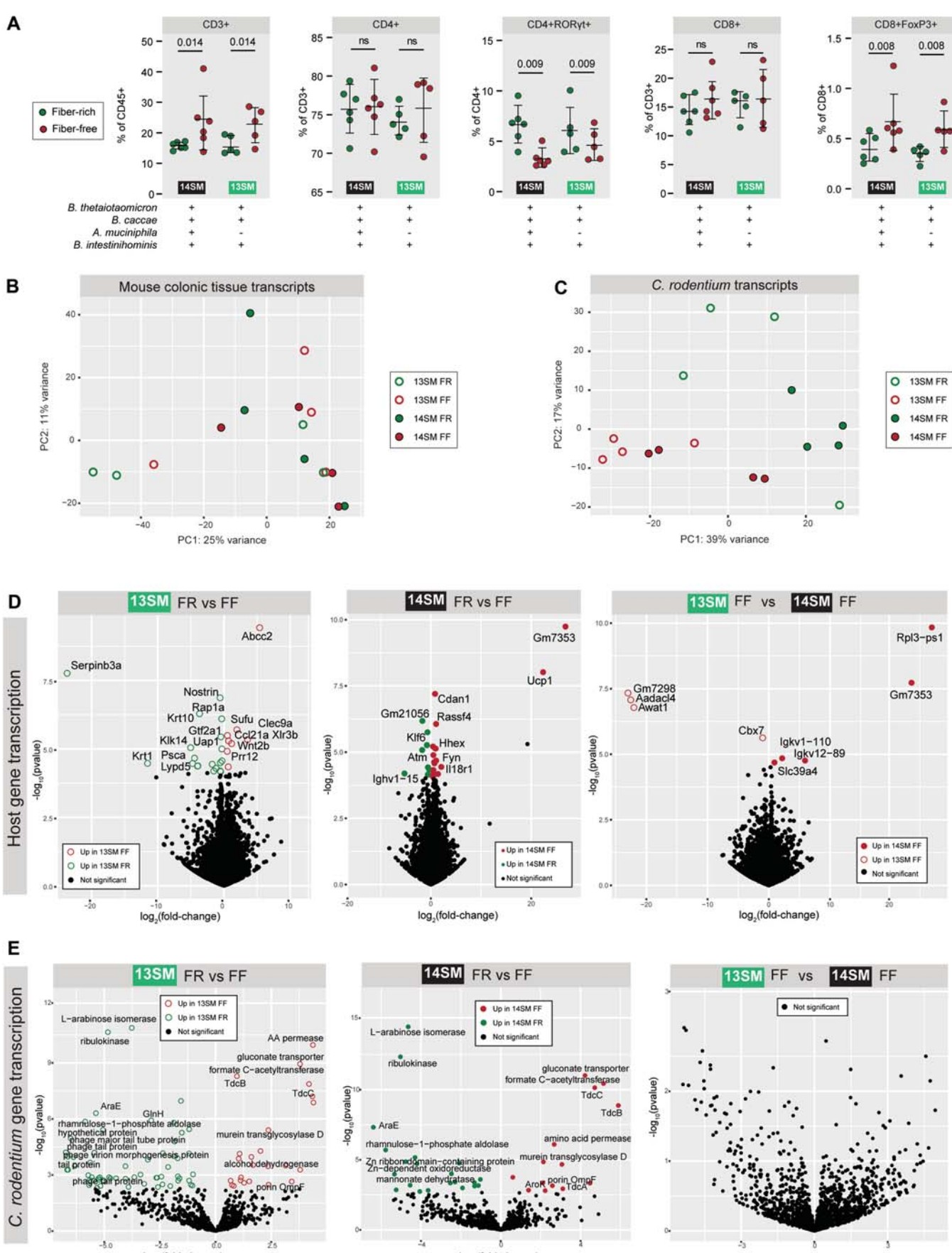

**Figure 3.  Exploration of host immune response and pathogen transcriptome to tease apart factors behind increased pathogen susceptibility.**

(A) Select immune cell populations as a percent of the parent population among mice at 3 DPI ($n = 5–6$ per group, no outliers), determined by fluorescence-activated cell sorting (FACS). See Fig. EV3B for the gating strategy. Population percentages were analyzed by two-way ANOVA (main effects only) with adjustment for multiple comparisons between SMs of the same diet (shown) and between diets of the same SM (see Source Data) using the Benjamini–Hochberg method. Error bars show SD and the center is the arithmetic mean. (B) Principal Components Analysis (PCA) plot of the mouse colonic tissue transcripts at 3 DPI separated by an experimental group; see Dataset EV2 for underlying data. (C) PCA plot of the *C. rodentium* transcripts at 3 DPI separated by an experimental group; see Dataset EV4 for underlying data. (D) Volcano plot of murine host gene transcription in colonic tissues at 3 DPI. Significance based on adjusted $p$ value <0.05 based on Wald test adjusted using the Benjamini–Hochberg method in DESeq2. Not all gene names are visible due to text overlaps; see Dataset EV3 for full results. (E) Volcano plot of *C. rodentium* gene transcription in cecal contents at 3 DPI. Significance based on adjusted $p$ value <0.05 based on Wald test adjusted using the Benjamini–Hochberg method in DESeq2. Not all gene names are visible due to text overlaps; see Dataset EV5 for full results. Source data are available online for this figure.

transcriptomic profiles offered a limited explanation as to why the FF-fed 14SM mice were more susceptible to *C. rodentium* infection.

Lacking evidence for altered susceptibility in the host transcriptome data, we next examined whether changes in the transcriptomic profile of *C. rodentium* (Dataset EV4) might provide greater insight. We identified numerous diet-specific changes in *C. rodentium* gene expression; however, the presence of *A. muciniphila* within the same FF diet background (13SM FF vs 14SM FF) had no effect on *C. rodentium* transcription (Fig. 3E; Dataset EV5). While there are some intriguing observations in the transcript data of *C. rodentium*, in particular, the transcription of a number of phage-type genes in the 13SM FR group, no transcripts of genes encoded on the locus of enterocyte effacement (LEE) pathogenicity island were altered in the *C. rodentium* transcriptome under any conditions (Dataset EV5).

In summary, our host immune and transcriptome, as well as pathogen transcriptome readouts, did not provide clear clues about how *A. muciniphila* could be altering susceptibility to *C. rodentium* in the context of the FF diet. Nevertheless, these results still hold importance because they clarify that the pathogen and the host respond the same way in 13SM FF and 14SM FF mice, despite resulting in strikingly different susceptibilities to the pathogen. Thus, these results point to the fact that the altered responses of *A. muciniphila* itself and also the accordingly altered responses of other community members likely play a critical role in enhancing susceptibility to *C. rodentium* in FF-fed mice.

## Fiber-deprived *A. muciniphila* increases mucus penetrability and alters community activity

We have previously shown that during fiber deprivation, the 14SM community is characterized by a thinner mucus layer (Desai et al, 2016). In 40-day FF-fed, uninfected mice, we validated this phenotype in the current study and further showed that there is a similar trend in the 13SM group, though the difference is more pronounced due to a generally higher mucus thickness among 13SM FR-fed mice (Fig. 4A). In addition to the thickness of the mucus layer, its integrity is important (Schroeder et al, 2018); thus, we used bacteria-sized beads (1 μm) to measure mucus penetrability (Fig. 4B). We observed an increase in the penetrability of the colonic mucus layer of the 14SM FF group compared to the FR group, which was significant by pairwise comparison ($p = 0.046$), although the adjusted $p$ value exceeded the traditional cutoff of 0.05 (Fig. 4B). However, there was no difference between the 13SM groups before or after correction for multiple comparisons (Fig. 4B), which suggests a potential mechanism by which *A. muciniphila*

could increase the *C. rodentium* susceptibility under FF conditions. It bears noting that the presence of *A. muciniphila* strongly influences the gut mucus layer; accordingly, the significant trend ($p = 0.046$) of the direct comparisons of the mucus penetrability between the FR and FF groups of a specific microbial composition (13SM or 14SM) should not be disregarded. Our results suggest alterations in mucus integrity that can stem from changes in host mucin secretion or glycosylation or from altered degradation by the commensal microbiome. The importance of the mucus layer integrity compared to its thickness is further highlighted by our prior work in germ-free mice, which, despite possessing a thin mucus layer, exhibit a diet-independent resistance toward *C. rodentium* infection comparable to 14SM FR-fed mice (Desai et al, 2016).

In order to assess whether the FF diet and the presence of *A. muciniphila* might be indirectly linked to *C. rodentium* susceptibility—i.e., through its influence on other community members—we performed pairwise correlation analyses on the abundances of each bacterium across 13SM and 14SM mice fed an FR or FF diet (Fig. EV4). Although we identified a number of statistically significant bacterial abundance co-associations that were altered between the two diets or between 13SM and 14SM colonized mice, we were unable to identify positive associations shared by both *A. muciniphila* and *C. rodentium* that were exclusively present in 14SM FF-fed mice and might thereby explain the increased susceptibility to infection. However, there was a significant negative correlation between *Marvinbryantia formatexigens* and both *A. muciniphila* ($r = −0.171$, adjusted $p = 0.03$) and *C. rodentium* ($r = −0.232$, adjusted $p < 0.01$), which was only observed within 14SM FF mice, suggesting that the acetogen *M. formatexigens* could be conferring some protection against infection under the FR and 13SM conditions.

To further investigate the role of *A. muciniphila*, we analyzed its microbial metatranscriptome data in the infected mice (same data from which *C. rodentium* transcripts were analyzed in Fig. 3). A 12-fold higher proportion of transcripts mapped to *A. muciniphila* on the FF diet, compared to on the FR diet (Fig. 4C). Focusing on mucin-targeting enzymes in particular, sialidase and α-*N*-acetylgalactosaminidase expression was significantly higher in FF mice compared to their FR counterparts (Fig. 4D). When comparing 14SM and 13SM FF-fed mice, however, the overall levels of transcripts mapping to mucin-targeting enzymes did not differ in the presence of *A. muciniphila*, as the niche was apparently taken up by mucin generalists such as *B. caccae* (Fig. 4D). Taking together, these results are important to support the conclusion that microbial over-foraging of the mucus layer alone is insufficient to explain the difference in *C. rodentium* susceptibility, as it is

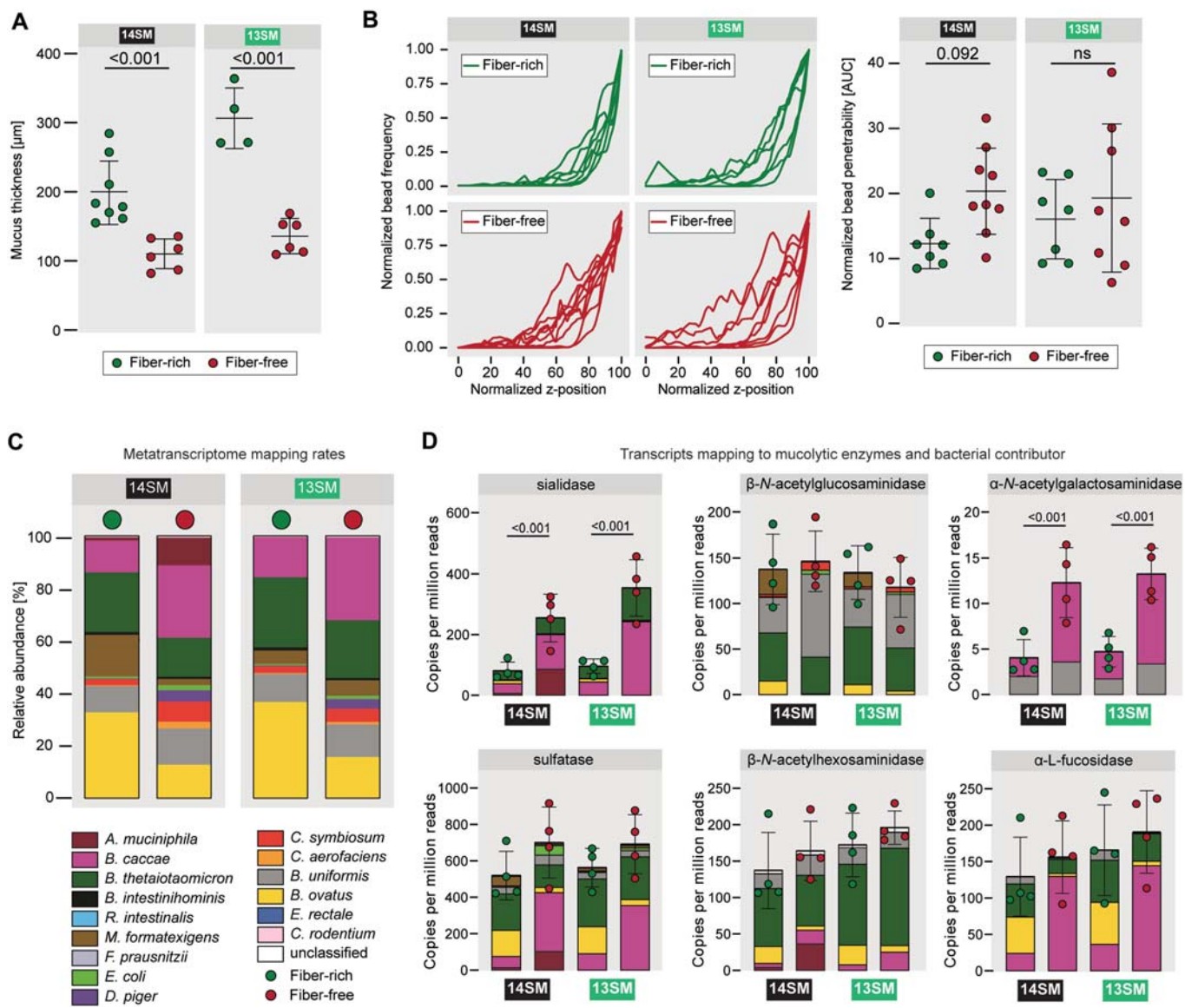

**Figure 4. Dietary fiber deprivation-induced erosion of the gut mucus layer partially explains increased *C. rodentium* susceptibility.**

(A) Mucus thickness of uninfected 13SM and 14SM mice on the fiber-rich (FR) and fiber-free (FF) diet based on Muc2 antibody staining ($n = 4$–8 per group). Error bars represent SD and the center of the arithmetic mean; two-way ANOVA (main effects only) with adjustment for multiple comparisons between SMs of the same diet (shown) and between diets of the same SM (see Source Data) using the Benjamini–Hochberg method. (B) Ex vivo mucus layer penetrability assessment on uninfected mice. The area under the curve (AUC) of the normalized bead penetrability was determined for each mouse ($n = 7$–9 per group following the removal of one outlier from 14SM FR). Error bars represent SD and the center of the arithmetic mean; two-way ANOVA (main effects only) with adjustment for multiple comparisons between SMs of the same diet (shown) and between diets of the same SM (see Source Data) using the Benjamini–Hochberg method. ns non-significant (adjusted $p > 0.1$). (C) Mapping rates of cecal RNA reads to the 14SM and *C. rodentium* genomes, averaged by diet and SM group. (D) Normalized counts (per million reads) of transcripts mapping to genes for mucin-targeting enzymes (sialidase, β-*N*-acetylglucosaminidase, β-*N*-acetylgalactosaminidase, sulfatase, β-*N*-acetylhexosaminidase, α-ʟ-fucosidase) according to UniRef90 names using HUMANn3 ($n = 4$ per group, no outliers). Error bars represent SD and the center of the arithmetic mean; two-way ANOVA (main effects only) with adjustment for multiple comparisons between SMs of the same diet and between diets of the same SM using the Benjamini–Hochberg method. Source data are available online for this figure.

observed even in the group that shows resistance to infection (13SM FF).

Given that the only clear difference between the groups stemmed from the presence or absence of *A. muciniphila*, we returned to the metatranscriptome to identify significant changes in transcription within this taxon. Among the 97 differentially abundant transcripts mapping to *A. muciniphila*

between FR and FF-fed mice, 21 are hypothetical proteins, that is, without a known function (Fig. 5A). Taking a more targeted approach, we noted that FF-fed mice exhibited an increase in transcripts that have been reported to have anti-inflammatory properties via induction of IL-10: UniRef90_B2UR41 or Amuc_1100 encoding a pili-like protein (Ottman et al, 2017) and UniRef90_A0A139TV36 encoding threonine-tRNA ligase (Kim

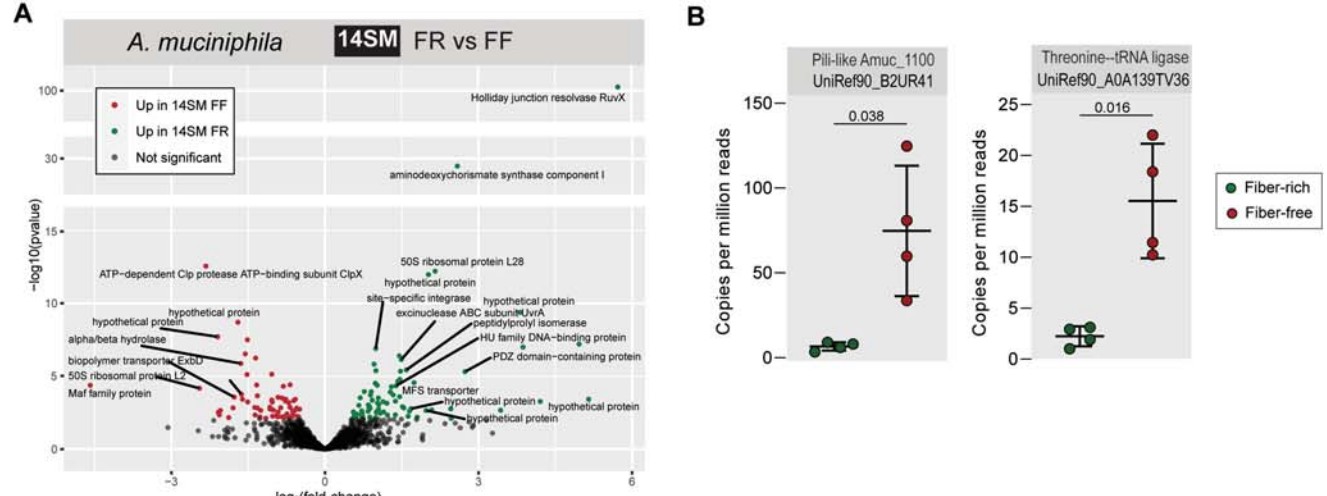

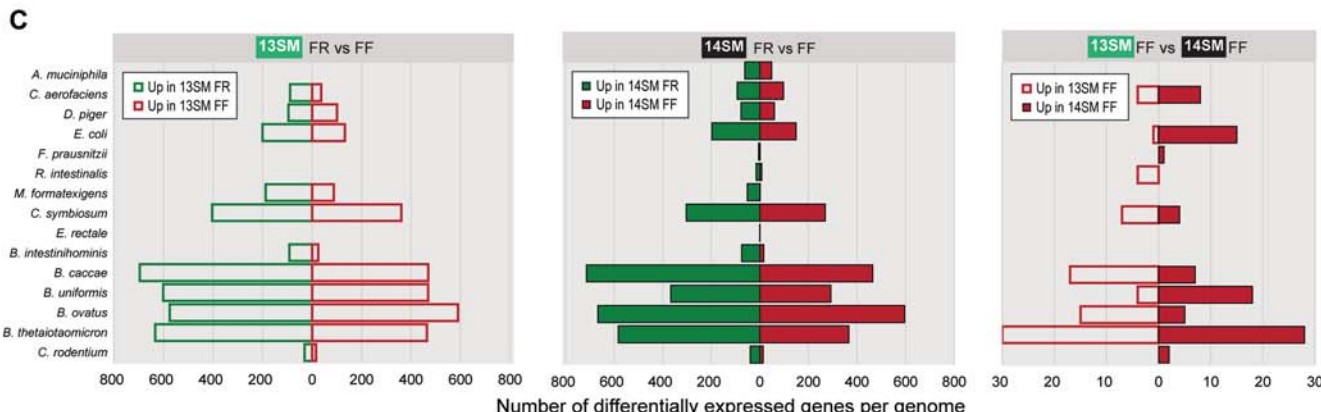

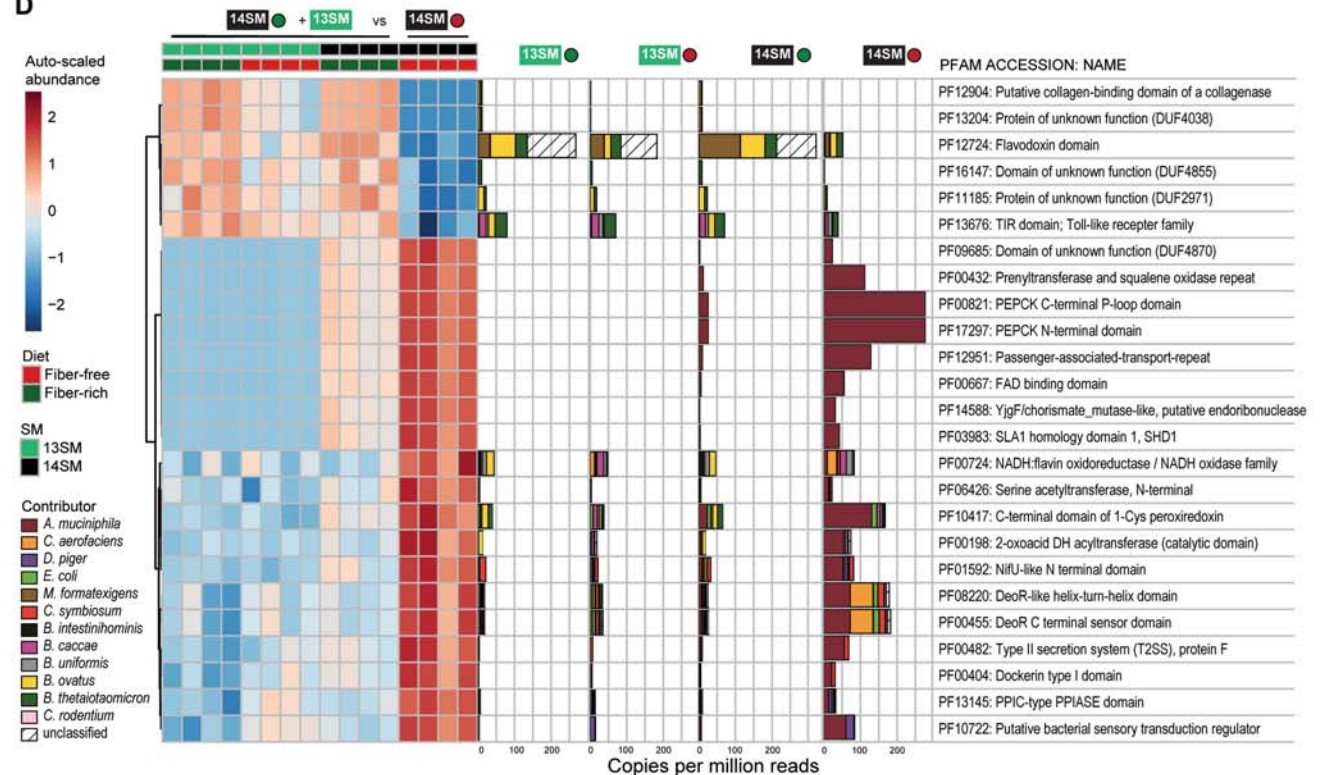

◀ **Figure 5.  Cecal metatranscriptomes highlight altered community activities and persistent functional knowledge gaps.**

(A) Volcano plot of *Akkermansia muciniphila* transcription in cecal contents at 3 days post-infection (DPI) in 14SM fiber-rich (FR) (*Citrobacter rodentium* resistant) and 14SM fiber-free (FF) (*C. rodentium* susceptible) mice, mapped using Salmon. Significance based on adjusted *p* value <0.05 based on Wald test adjusted using the Benjamini–Hochberg method in DESeq2. (B) Dotplot of known immunomodulatory gene products transcribed by *A. muciniphila* in cecal contents of 14SM FR and 14SM FF mice at 3 DPI based on UniRef90 identifiers in HUMANn3 based on UniRef90 identifiers. Left: Pili-like membrane protein Amuc_1100. Right: threonine-tRNA ligase. Error bars represent SD and the center is the arithmetic mean. Copies per million reads attributed to each transcript, *p* values calculated using Welch's *t*-test. (C) Barplot indicating the number of differentially expressed genes using DESeq2 in the indicated group-wise comparison for each member of the 14SM community plus *C. rodentium*. Transcripts mapped using Salmon, with significance based on adjusted *p* value <0.05 based on Wald test adjusted using the Benjamini–Hochberg method in DESeq2. (D) Left: Heatmap of the top 25 (sorted by lowest adjusted *p* value based on Wald test adjusted using Benjamini–Hochberg method in DESeq2, all <0.05) differentially expressed genes according to Pfam identifier, regrouped from UniRef IDs in HUMANn3, for all *C. rodentium* resistant groups (14SM FR, 13SM FR, and 13SM FF) compared to the susceptible condition (14SM FF). Abundances are auto-scaled (mean-centered and divided by standard deviation). Right: Copies per million reads attributed to each Pfam, colored by bacterial contributor, in each diet and SM group. See Dataset EV6 for underlying data. For all panels, $n = 4$ per group, no outliers. Source data are available online for this figure.

et al, 2023) (Fig. 5B). While these are promising microbial-derived factors in the treatment of autoimmune diseases, it is possible that their immunosuppressive effects are deleterious in the context of enteropathogenic infection.

Finally, as elegantly demonstrated by Weiss et al (Weiss et al, 2023), bacterial community dynamics are highly dependent on both the nutritional conditions and the presence or absence of other key consortium members. Overall, the total number of differentially expressed genes (DEGs) was lower between 13SM and 14SM mice fed the FF diet, than between the same SM fed the FR diet, highlighting the strong impact of diet on the activity of the overall community (Fig. 5C). Among FF-fed mice, *Bacteroides* spp., *C. symbiosum*, *C. aerofaciens*, and *E. coli* appeared most responsive to the absence of *A. muciniphila*, as detailed in Fig. EV5. A limitation of this analysis is that each locus in the genome is treated independently, even if they encode the same product. For example, transcripts corresponding to TonB-dependent receptors are simultaneously differentially expressed in 13SM FF and 14SM FF because they correspond to different loci in the *B. ovatus* genome (Fig. EV5). Therefore, in order to calculate differential expression according to the unique gene product, we used HUMANn3 to map the reads to a custom database of the 14SM plus *C. rodentium* and regrouped the unstratified output to the Pfam database (Mistry et al, 2021) for consistent nomenclature (Dataset EV6). Again, differentially expressed gene products between the non-susceptible and susceptible (14SM FF) phenotypes were identified based on the unstratified data and then visualized according to the stratified output (Fig. 5D). Here, we can see the altered transcriptional profiles in greater detail, particularly the contribution of transcripts from *A. muciniphila*, and note that many of the protein families correspond to functions to combat oxidative stress (PF00667, PF00724, PF10417), gluconeogenesis (PF00821, PF17297), or are involved in protein secretion (PF12951, PF00482) (Fig. 5D).

The increased susceptibility to *C. rodentium* infection observed in 14SM FF-fed mice cannot be attributed to either the diet or microbiome alone, but rather depends on both factors to produce a local environment that supports infection (Fig. 6). The host and pathogen both behave as expected—mounting an innate immune response against infection or expressing virulence factors to establish infection, respectively—regardless of the diet or SM. The decisive factor determining the outcome of this battle rests in the microbiome, which, unlike the host or pathogen, collectively alters their gene expression in a diet- and microbiome-dependent manner. Ultimately, in leveraging a strain dropout approach, we find that a single mucin-degrading bacterium, *A. muciniphila*, plays a pivotal role in modulating pathogen susceptibility in a diet-dependent manner (Fig. 6).

## Discussion

Dietary fiber consumption is well known to confer a myriad of health benefits (Sonnenburg and Sonnenburg, 2014; Makki et al, 2018; Gill et al, 2021; Wolter et al, 2021a). Moreover, fiber consumption is indispensable to avoid a functional shift of the gut microbiota toward mucin degradation, which erodes the gut mucus layer and increases pathogen susceptibility (Desai et al, 2016; Neumann et al, 2021). By manipulating the presence of functionally relevant members of the microbiota, our work offers insights into the multifactorial and ecological nature of susceptibility to enteropathogenic infection, with the key finding that *A. muciniphila* increases susceptibility to infection in concert with other members of the microbiome, but only in the context of a fiber-deficient diet (Figs. 2,6). Importantly, adding *A. muciniphila* as the sole mucin degrader to the synthetic microbiota appeared to be protective to *C. rodentium* colonization in the context of a fiber-sufficient diet. However, we also show that resistance to infection can be achieved on a fiber-deprived diet through the removal of *A. muciniphila*, which contributes to deterioration of the gut mucus layer (Fig. 4B) and alters the activity of other gut community members (Fig. 4C,D). Our study highlights the power of a strain dropout approach within a defined consortium to elucidate possible mechanistic links in the microbiota–pathogen–host axis.

Intriguingly, a fiber-deprived microbiota-led reduced mucus barrier alone is insufficient to heighten susceptibility to an enteric pathogen, but the presence of a microbial biomarker species—that is, a species whose presence or activity can be used to predict susceptibility to disease or disease-course—is essential. In this study, *A. muciniphila* can be considered a biomarker species since its presence or absence under fiber-deprived dietary conditions determines the course of *C. rodentium* infection. A fiber-deprived community harboring *A. muciniphila* exhibits increased penetrability of the mucus barrier, thereby likely increasing the influx of microbial antigens that might aid in the heightened susceptibility. Furthermore, the beneficial aspects of mucolytic bacteria are also evident, as we noticed increased markers of pathogen susceptibility in the absence of all four mucolytic bacteria, possibly owing to immature mucus barrier function in the complete absence of mucolytic bacteria.

   

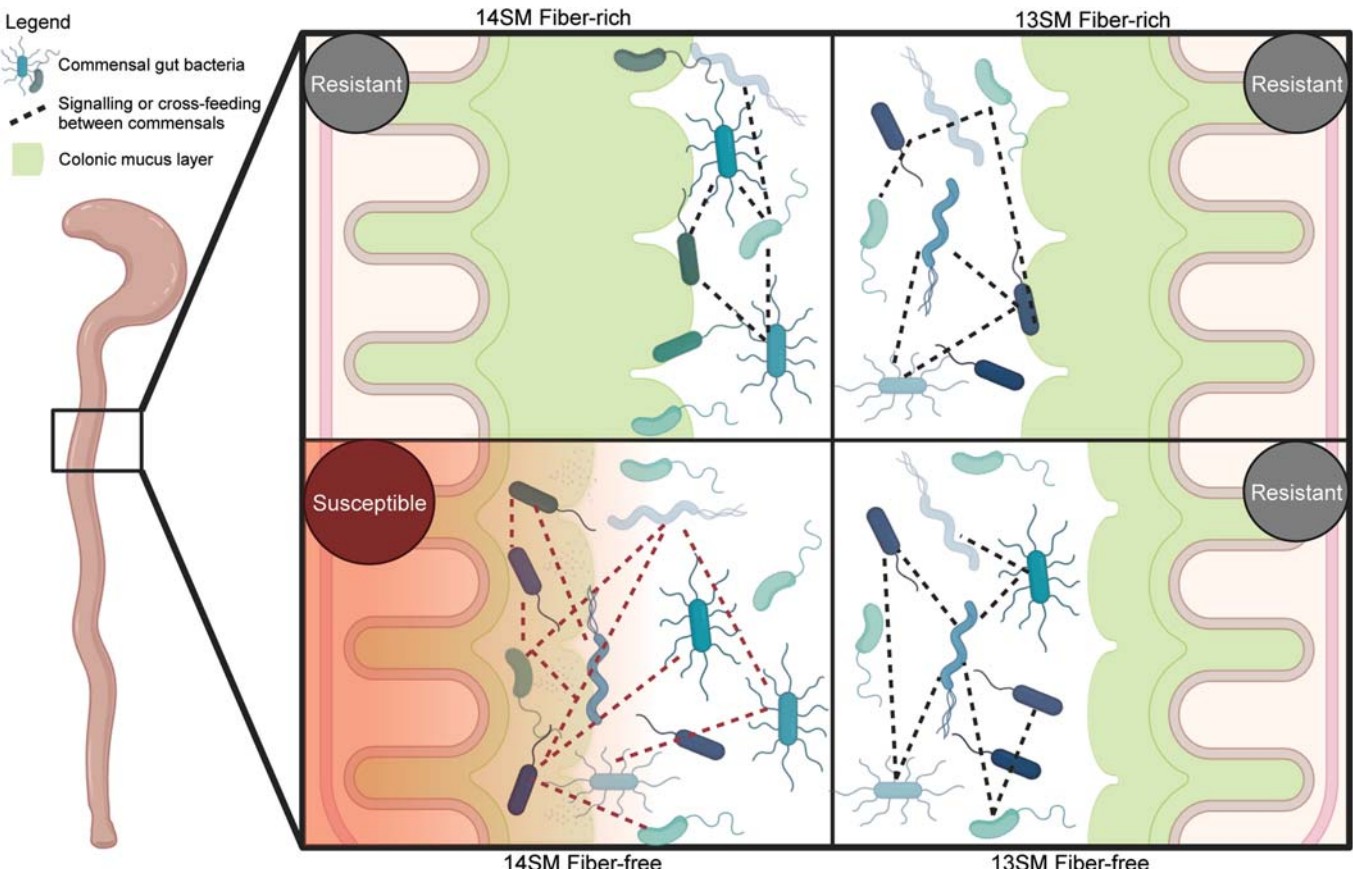

**Figure 6. Impaired mucosal integrity and altered community dynamics underlie observed microbiota and diet-dependent susceptibility to *Citrobacter rodentium* infection.**

Visual illustration of the impact of mucin-degrading bacterium *A. muciniphila* in mice fed a fiber-deprived diet on *C. rodentium* susceptibility. While host immune response and pathogen activities are unchanged in these conditions, a compromised mucosal barrier (illustrated by a thinner colonic mucus barrier and increased mucus permeability under fiber-free conditions in 14SM) and altered metatranscriptional profile reflecting changes in interactions between commensal gut bacteria (illustrated by dashed red lines between bacteria in fiber-deprived 14SM) suggest that a combination of these factors contributes to the observed disease phenotype. 14SM 14-member synthetic microbiota, 13SM 13-member synthetic microbiota (lacking *Akkermansia muciniphila*). Created using BioRender.com.

As *A. muciniphila* is regarded as a potential probiotic bacterium (Zhai et al, 2019; Ashrafian et al, 2021; Cani et al, 2022; Daniel et al, 2023), and is connected even to extra-intestinal autoimmune diseases (Miyauchi et al, 2023), the effects of this bacterium on the activity of other members of the gut microbial community and under different dietary regimens must be considered when designing probiotic therapies using this microbe. Indeed Mao et al, show that treating mice with glycosaminoglycan polymer hyaluronan on a normal chow impacts the abundance of various members of the microbiome, enriching *A. muciniphila* in particular, which in turn confers protection against *C. rodentium* infection (Mao et al, 2021). The opposing and intriguing observation that *A. muciniphila* can reduce the pathogen load in the presence of a fiber-sufficient diet and can exacerbate pathogen susceptibility in the absence of fiber, underscores the nuanced and widely overlooked concept that, depending on the dietary context, this bacterium can be either beneficial or detrimental. This observation echoes the views of some other researchers that *A. muciniphila*'s impact on health need to be rationally considered (Cirstea et al, 2018; Luo et al, 2022). Our microbial transcriptome

analyses highlight that *A. muciniphila* behaves differently on two diets, with many differentially transcribed genes encoding hypothetical proteins. A key question that remains unanswered is which of these differentially expressed gene products play an important role in altering the pathogen susceptibility, which relies in part on increasing knowledge of the structure and function of these hypothetical proteins.

A related aspect for future studies is to understand the connection between increased mucus penetrability and differentially transcribed genes of *A. muciniphila*. One possible explanation is that the increased expression of pili-like proteins (Ottman et al, 2017) and threonine—tRNA ligase (Kim et al, 2023) by *A. muciniphila* under fiber-deprived conditions might tune the immune system in a way that increases vulnerability to the pathogen. In the case of the aforementioned components, IL-10 induction might represent a deleterious immune response that ultimately contributes to the exacerbated pathogen susceptibility. Differences in the microbial compositions of laboratory mice can impact susceptibility to *C. rodentium* infection through altered production of SCFAs (Osbelt et al, 2020). Since we observed

reduced propionate—an SCFA produced by *A. muciniphila* (Derrien et al, 2004)—in *A. muciniphila*-containing FF-fed mice, it is possible that the altered metabolism of fiber-deprived *A. muciniphila* contributes to increased pathogen susceptibility. Altogether, the molecular mechanisms through which fiber-deprived *A. muciniphila* alters susceptibility to *C. rodentium* could involve multiple independent contributors.

The next step to precisely decipher the contribution of individual molecular mechanisms, of how *A. muciniphila* modulates disease susceptibility in a diet-dependent manner, includes the implementation of genetic systems, for example, by using transposon mutants, as introduced by Davey et al, (Davey et al, 2023). Using such mutants might also help to understand which specific responses of *A. muciniphila* majorly impact the broader community dynamics, especially considering that the degradation of the gut mucus by mucolytic bacteria releases glycan residues that can be metabolized by either the mucolytic bacteria themselves or other members of the gut microbiota (Belzer et al, 2017; Crost et al, 2018; Raimondi et al, 2021). Changes in which mucosal sugars are released in the presence or absence of *A. muciniphila* under fiber-deprived conditions could also play a role in the susceptibility, as it was recently shown that sialic acid—a sugar present in mucin glycoproteins—plays a vital role in aiding the transition of *C. rodentium* from lumen to mucosa (Liang et al, 2023). Although we did not measure free sialic acid directly, we detected an increase in sialidase transcripts among mice fed an FF diet (Fig. 4D), suggesting that more sialic acid might be available as a result of altered microbial activities under fiber-deprived conditions. Furthermore, *A. muciniphila* can release sialic acid during the degradation of mucin, yet its inability to subsist on sialic acid alone suggests that *A. muciniphila* does not use this compound for growth (Shuoker et al, 2023). It is, therefore, plausible that the combination of an FF diet and the presence of *A. muciniphila* results in the highest sialic acid availability and thereby facilitates the invasion of *C. rodentium*, which should be verified in future experiments. Another recent study showed that dietary restriction of L-serine enhances mucin degradation by *A. muciniphila* and promotes encroachment of adherent-invasive *Escherichia coli* (AIEC) to the epithelial niche, where the pathogen acquires host L-serine from the epithelium to proliferate (Sugihara et al, 2022). Future studies are needed to verify possible metabolic interactions between *A. muciniphila* and *C. rodentium*.

In line with a previous study that showed how a high-fat diet impacts susceptibility to *C. rodentium* (An et al, 2021), our study highlights how microbial ecology and the presence or absence of certain key taxa impact disease susceptibility by altered dietary habits. The role of diet in this dynamic is particularly relevant given that dietary fiber intake in many industrialized countries remains below the recommended intake for adults of 25 g/day (European Food Safety Authority, 2010). It is likely that *A. muciniphila* is just one of the potential biomarkers that can be used to predict susceptibility to enteric pathogen infection. Identifying additional such biomarker species and understanding how their role in disease susceptibility can be modified through the diet could prove to be an important tool to help lower the burden of human foodborne enteropathogenic infections—a looming challenge in the face of the dual threats of altered food supply systems and antimicrobial resistance (Willett et al, 2019).

# Methods

## Reagents and tools table

| Reagent/Resource | Reference or source | Identifier or catalog number |
|---|---|---|
| **Experimental Models** | | |
| Swiss Webster (*Mus musculus*), male and female, 6–8 weeks old | Taconic Biosciences | Tac:SW |
| *Akkermansia muciniphila*: DSM 22959, type strain | DSMZ | Cat # DSM 22959 |
| *Bacteroides caccae*: DSM 19024, type strain | DSMZ | Cat # DSM 19024 |
| *Bacteroides ovatus*: DSM 1896, type strain | DSMZ | Cat # DSM 1896 |
| *Bacteroides thetaiotaomicron*: DSM 2079, type strain | DSMZ | Cat # DSM 2079 |
| *Bacteroides uniformis*: ATCC 8492, type strain | ATCC | Cat # ATCC 8492 |
| *Barnesiella intestinihominis*: YIT11860 | DSMZ | Cat # DSM 21032 |
| *Clostridium symbiosum*: DSM 934, type strain, 2 | DSMZ | Cat # DSM 934 |
| *Collinsella aerofaciens*: DSM 3979, type strain | DSMZ | Cat # DSM 3979 |
| *Desulfovibrio piger*: ATCC 29098, type strain | ATCC | Cat # ATCC 29098 |
| *Escherichia coli* HS | ATCC | N/A |
| *Eubacterium rectale*: DSM 17629, A1-86 | DSMZ | Cat # DSM 17629 |
| *Faecalibacterium prausnitzii*: DSM 17677, A2-165 | DSMZ | Cat # DSM 17677 |
| *Marvinbryantia formatexigens*: DSM 14469, type strain, I-52 | DSMZ | Cat # DSM 14469 |
| *Roseburia intestinalis*: DSM 14610, type strain, L1-82 | DSMZ | Cat # DSM 14610 |
| **Antibodies** | | |
| Rat anti-mouse CD16/CD32 (Mouse BD Fc Block™) | BD Biosciences | Cat # 553141 RRID: AB_394656 |
| BV605-conjugated rat anti-mouse CD4, monoclonal RM4-5, 1:700 | Biolegend | Cat # 100548 RRID: AB_2563054 |
| BV650-conjugated rat anti-mouse CD45R/B220, monoclonal RA3-6B2, 1:88 | BD Biosciences | Cat # 563893 RRID: AB_2738471 |
| BV711-conjugated rat anti-mouse CD3, monoclonal 17A2, 1:88 | Biolegend | Cat # 100241 RRID: AB_2563945 |
| BV780-conjugated rat anti-mouse CD45, monoclonal 30-F11, 1:88 | BD Biosciences | Cat # 564225 RRID: AB_2716861 |
| FITC-conjugated rat anti-mouse CD335/NKp46, monoclonal 29A1.4, 1:100 | Biolegend | Cat # 137606 RRID: AB_2298210 |
| PE-Cy5-conjugated rat anti-mouse CD8, monoclonal 53-6.7, 1:700 | Biolegend | Cat # 100710 RRID: AB_312749 |

| Reagent/Resource | Reference or source | Identifier or catalog number |
|---|---|---|
| eFluor™ 450-conjugated rat anti-mouse FoxP3, monoclonal FJK-16s, 1:200 | Invitrogen eBioscience™ | Cat # 48-5773-82 RRID: AB_1518812 |
| PE-conjugated mouse anti-mouse GATA3, monoclonal 16E10A23, 1:44 | Biolegend | Cat # 653804 RRID: AB_2562723 |
| PE-eFluor™ 610-conjugated rat anti-mouse EOMES, monoclonal Dan11mag, 1:100 | Invitrogen eBioscience™ | Cat # 61-4875-82 RRID: AB_2574614 |
| PE-Cy7-conjugated mouse anti-mouse Tbet, monoclonal 4B10, 1:44 | Biolegend | Cat # 644824 RRID: AB_2561761 |
| APC-conjugated rat anti-mouse RORγt, monoclonal AFKJS-9, 1:22 | Invitrogen eBioscience™ | Cat # 17-6988-82 RRID: AB_10609207 |
| **Oligonucleotides and sequence-based reagents** | | |
| Synthetic microbiota species primers | Desai et al, 2016 | N/A |
| V4 16S rRNA gene primers | Kozich et al, 2013 | N/A |
| **Chemicals, enzymes and other reagents** | | |
| Zombie NIR ™ Fixable Viability Kit | Biolegend | Cat # 423106 |
| Invitrogen™ UltraPure™ DNase/RNase-free distilled water | Fisher Scientific | Cat # 10358742 |
| LB Agar (Luria/Miller) | Carl Roth | Cat # X969.1 |
| Tryptone yeast extract glucose (TYG) | Desai et al, 2016 | N/A |
| Modified Baar's medium for sulfate reducers | Desai et al, 2016 | N/A |
| Custom chopped meat broth | Desai et al, 2016 | N/A |
| Modified yeast extract, casitone, and fatty acid (YCFA) medium (USA version) | Desai et al, 2016 | N/A |
| Modified yeast extract, casitone, and fatty acid (mYCFA) medium (Luxembourg version) | Steimle et al, 2021 | N/A |
| Hanks' buffered saline solution (HBSS) with phenol red and without calcium and magnesium | Lonza | Cat # 10-543F |
| Methanol | Carl Roth | Cat # 4627.6 |
| Acetic acid 100% | Carl Roth | Cat # 6755.1 |
| Phenol/Chloroform/Isoamyl alcohol (25:24:1) pH 8.0 | Fisher Scientific | Cat # 10535691 |
| Glass Beads, acid-washed | Sigma Aldrich | Cat # G1277 |
| Sodium dodecyl-sulfate | Fisher Scientific | Cat # BP166 |
| Chloroform | Fisher Scientific | Cat # 10122190 |
| Isopropanol | Fisher Scientific | Cat # 10477070 |
| Ethanol | VWR | Cat#1.08543.0250 |
| DNeasy Blood & Tissue Kit | QIAGEN | Cat # 69506 |
| RNAprotect™ Tissue Reagent | QIAGEN | Cat # 76104 |
| TRIzol™ Reagent | Invitrogen | Cat # 15596018 |
| DNase I, RNase-free | Thermo Scientific™ | Cat # EN0521 |

| Reagent/Resource | Reference or source | Identifier or catalog number |
|---|---|---|
| Phenol/Chloroform/Isoamyl Alcohol (125:24:1) pH 4.3 | Fisher BioReagents | Cat # 10699543 |
| RNAProtect™ Bacteria Reagent | QIAGEN | Cat # 76506 |
| RNeasy Mini Kit | QIAGEN | Cat # 74104 |
| SYBR™ Green I Nucleic Acid Gel Stain, 10,000X concentrate in DMSO | Invitrogen | Cat # 10710004 |
| dNTP Set (100 mM) Solution | Invitrogen | Cat # 10083252 |
| SYTO-9 Green Fluorescent Nucleic Acid Stain | Invitrogen | Cat # S34854 |
| FluoSpheres™ carboxylate beads (1 μm, red 580/605) | Invitrogen | Cat # F8821 |
| **Software** | | |
| ImageJ | https://imagej.net | N/A |
| SoftMax Pro 7 Software | Molecular Devices | N/A |
| CFX Maestro 1.1 v4.1.2433.1219 | Biorad | N/A |
| NovoExpress v1.6.2 | Agilent | N/A |
| FlowJo v10.8.1 | BD Biosciences | N/A |
| Prism v10.1.2 | GraphPad Software, Inc., San Diego, CA, USA | N/A |
| mothur v1.44.1 | Schloss et al, 2009 | N/A |
| salmon v1.6.0 | Patro et al, 2017 | N/A |
| DESeq2 v1.30.1 | Love et al, 2014 | N/A |
| humann v3.1.1 | Beghini et al, 2021 | N/A |
| kneaddata | https://github.com/biobakery/kneaddata | N/A |
| Zen 3.0 | Carl Zeiss Microscopy GmbH | N/A |
| Imaris | Oxford Instruments Imaris | N/A |
| BacSpace | Earle et al, 2015 | N/A |
| **Other** | | |
| Standard lab chow (fiber-rich, FR) | LabDiet | Cat # 5013 |
| Acetylated galactoglucomannan (AcGGM) diet | Envigo-Teklad | N/A |
| Fiber-free (FF) diet based on the Harlan.TD08810 (Luxembourg version) | SAFE Diets | N/A |
| Fiber-free (FF) diet based on the Harlan.TD08810 (USA version) | Envigo-Teklad | Cat # TD.130343 |
| Anaerobic chamber | Coy manufacturing | Vinyl Type A + Type B |
| Stainless Steel Beads, 5 mm | QIAGEN | Cat # 69989 |
| RETSCH MM 400 | Fisher Scientific | Cat # 10573034 |
| PTFE Adapter Rack | Fisher Scientific | Cat # 10122852 |
| Implen NanoPhotometer N60 Micro-Volume UV-VIS Spectrophotometer | Fisher Scientific | Cat # 15442203 |

| Reagent/Resource | Reference or source | Identifier or catalog number |
|---|---|---|
| CFX96TM RealTime System (C1000 Thermal Cycler) | Biorad | Cat # 1855195 |
| Quick-16S™ NGS Library Prep Kit | BaseClear | Cat # D6400 |
| MiSeq Reagent Kit v2 | Illumina | Cat # MS-102-2003 |
| Illumina MiSeq | Illumina | Cat # SY-410-1003 |
| Agilent 2100 Bioanalyzer | Agilent | Cat # AG-2100 |
| Agilent RNA 6000 Nano Kit | Agilent | Cat # 5067-1511 |
| Illumina Stranded Total RNA Prep with Ribo-Zero Plus kit | Illumina | Cat # 20040529 |
| NovaSeq 6000 SP Reagent Kit v1.5 | Illumina | Cat # 20028402 |
| Illumina NovaSeq 6000 | Illumina | Cat # 20012850 |
| Perfusion chambers for bead assay | Gustafsson et al, 2012 | N/A |
| SpectraMax ABS Plus UV/Vis Single-Mode | Molecular Devices | Cat#5055292 |
| DuoSet mouse lipocalin-2/NGAL ELISA kit | R&D Biosystems | Cat # DY1857 |
| Lamina Propria Dissociation Kit for mice | Miltenyi Biotec | Cat # 130-097-410 |
| gentleMACS dissociator | Miltenyi Biotec | Cat # 130-093-235 |
| FOXP3/Transcription Factor Staining Buffer kit | Invitrogen eBioscience™ | Cat # 00-5523-00 |
| NovoCyte Quanteon flow cytometer | Agilent / ACEA Biosciences | Cat # 2010097 |
| Zeiss Apotome | Carl Zeiss Microscopy GmbH | N/A |
| Axio Observer Z1 | Carl Zeiss Microscopy GmbH | Cat # 491406-9880-010 |

## Ethical statement

Animal experiments in the United States were approved by the University of Michigan Institutional Animal Care & Use Committee. Animal experiments in Luxembourg were approved by both the University of Luxembourg Animal Experimentation Ethics Committee and by the Luxembourgish Ministry of Agriculture, Viticulture, and Rural Development (authorization no. LUPA 2019/52) and carried out according to the "Règlement Grand-Ducal du 11 janvier 2013 relatif à la protection des animaux utilisés à des fins scientifiques", according to Directive 2010/63/EU on the protection of animals used for scientific purposes.

## Animal diets

In both animal facilities, the fiber-rich (FR) diet was an autoclaved rodent chow (LabDiet #5013, St. Louis, MO, USA). For experiments in the United States, the fiber-free (FF) diet was manufactured and irradiated by Envigo (TD.130343, Indianapolis, IN, USA) and is based on the Harlan.TD08810 diet (Envigo, Indianapolis, IN, USA) (Desai et al, 2016). The fiber-free diet used in the Luxembourg

facility was manufactured and irradiated by SAFE Diets (Augy, France) based on the TD.130343 diet formulation used in the United States. The FF diet lacks dietary fiber, which has been compensated by an increase in glucose. The FF diet also contains crystalline cellulose, a polysaccharide that cannot be degraded by any member of the SM. The AcGGM diet was a modified version of the TD.130343 diet, manufactured and irradiated by Envigo (Indianapolis, IN, USA), which was supplemented with 7.5% of the dietary fiber acetylated galactoglucomannan (AcGGM).

## Experimental design

Germ-free (GF), 6- to 8-week-old, age-matched, male and female Swiss Webster mice were housed in iso-cages with a maximum of five animals of the same sex per cage. Light cycles consisted of 12 h of light and sterile water, with diets provided ad libitum. The experiments were performed in two different facilities, one facility in Luxembourg and the other in the United States. Results were reproduced independently of the location of the facility. Researchers were blinded to the treatment group for histological disease scoring, SCFA quantification, and FACS analysis. Prior to and 14 days following the initial gavage, all mice were maintained on the FR diet. Two weeks after the gavage, half of the mice per SM group were switched to the FF diet. Mice were maintained on their respective diets for up to 40 days, and fecal samples were collected as represented in Fig. 1A. One group of mice was kept on the FF diet for 5 days ($n = 4$) or 20 days ($n = 3$), respectively, to determine the timeline to develop a susceptible phenotype (Fig. EV2E). Aside from the shortened feeding period, these mice were treated identically to the mice under the 40-day feeding regimen. At the end of the feeding period, mice were either euthanized for pre-infection readouts or infected with ~$10^9$ CFUs of C. rodentium as previously described (Desai et al, 2016). Mice were observed daily following infection. Mice dedicated for FACS and host/microbial transcriptomics analyses were euthanized 3 days post-infection (DPI), while the remaining mice were observed for up to 10 DPI before euthanasia. In the United States, mice were euthanized by $CO_2$ asphyxiation followed by cervical dislocation, while mice in Luxembourg were euthanized directly by cervical dislocation. Colons were excised and either processed for bead penetration assay measurements or stored in Methacarn fixative for histological assessment. For FACS analysis in a subset of mice, colons were excised and processed, as described in the section "Immune cell profiling of colonic lamina propria". For RNA extraction, mesenteric lymph nodes and 2 mm sections of colonic tissue were stored in 1 ml RNAprotect Tissue Reagent for up to 2 weeks, then the RNAprotect was removed and the tissue stored at −80 °C until subsequent processing. Cecal contents were flash-frozen prior to storage at −80 °C for RNA extraction or SCFA measurements. Mice fed the AcGGM diet followed the same protocol as the mice fed the FF diet.

## Cultivation and administration of the SM

All SM-constituent strains were cultured and intragastrically gavaged into germ-free mice, as previously described (Steimle et al, 2021; Desai et al, 2016). For the experiment in the United States, 14SM bacteria were cultured individually in modified yeast extract, casitone and fatty acid (YCFA) medium (Roseburia

*intestinalis*, *Faecalibacterium prausnitzii*, and *Marvinbryantia formatexigens*), tryptone yeast extract glucose (TYG) medium (*Collinsella aerofaciens*), modified Baar's medium for sulfate reducers (*Desulfovibrio piger*) or custom chopped meat broth (*Bacteroides caccae*, *Bacteroides thetaiotaomicron*, *Bacteroides ovatus*, *Bacteroides uniformis*, *Barnesiella intestinihominis*, *Eubacterium rectale*, *Clostridium symbiosum*, *Escherichia coli*, and *Akkermansia muciniphila*) (Desai et al, 2016). For the experiments in Luxembourg, all 14SM bacterial cultures were grown individually in a further modified yeast extract, casitone, and fatty acid medium (mYCFA) (Steimle et al, 2021). Mice were intragastrically gavaged with up to 0.2 ml of one of the five synthetic human gut microbiota (SM) combinations on three consecutive days. The bacterial mixtures were freshly prepared with approximately equal volume of each strain (individual cultures ranged from $OD_{600}$ of 0.5–1.0). Selected strains were not included in the gavage mix depending on the specific SM (Fig. 1B).

## Colon length and mucus thickness measurements

Colon length was measured by taking pictures of the colons in their histology cassettes, followed by length measurements with ImageJ (https://imagej.net/) using the cassette size as reference. The mucus layer staining was performed as described previously (Parrish et al. 2023). Measurement of the colonic mucus layer thickness was performed as previously described, with cross-sectional images analysed in BacSpace (Earle et al, 2015).

## Histological disease scoring

Histological disease scoring was performed in a blinded fashion by Prof. Kathryn A. Eaton from the University of Michigan Medical School according to a modified version of the protocol by Meira et al (Meira et al, 2008). Briefly, scores were assigned based on inflammation, epithelial damage, hyperplasia/dysplasia, and submucosal edema.

## Ex vivo mucus layer penetrability assessment

Gut mucus layer penetrability was measured using fluorescent beads, as previously described (Gustafsson et al, 2012; Schroeder et al, 2018). Briefly, the colons were removed from euthanized mice and gently flushed using oxygenated ice-cold Krebs buffer (116 mM NaCl, 1.3 mM $CaCl_2 \times 2H_2O$, 3.6 mM KCl, 1.4 mM $KH_2PO_4$, 23 mM $NaHCO_3$, and 1.2 mM $MgSO_4 \times 7H_2O$). The muscle layer was removed by blunt microdissection while keeping colon tissue suspended in oxygenated, ice-cold Krebs buffer. Then, the distal mucosa was mounted in a perfusion chamber. The apical chamber contained oxygenated, ice-cold Krebs-mannitol buffer (Krebs buffer with 10 mM mannitol, 5.7 mM sodium pyruvate, and 5.1 mM sodium-L-glutamate), while the basolateral chamber contained oxygenated, ice-cold Krebs-glucose buffer (Krebs buffer with 10 mM glucose, 5.7 mM sodium pyruvate, and 5.1 mM sodium-L-glutamate) with 0.6 µg/ml SYTO-9. After a 10 min incubation in the dark, at room temperature, FluoSpheres™ carboxylate beads (1 µm, red 580/605, Invitrogen) were added apically and allowed to sediment on the tissue for 5 min. Next, the apical chamber was gently flushed with Krebs-mannitol buffer to remove excess beads and the tissue was incubated for an additional

10 min before visualizing with a microscope. For each tissue, 4–7 confocal images were taken as XY stacks with 5 µm intervals, from the epithelium to the beads. Mucus penetrability was calculated according to the distance of each bead to the epithelium.

## Nucleic acid extraction

Bacterial DNA was extracted from fecal pellets by phenol-chloroform extraction, as previously described (Steimle et al, 2021). Total RNA was extracted from the host mesenteric lymph nodes, colonic tissue, or cecal contents at 3 DPI, as previously described (Parrish et al, 2022b; Grant et al, 2023). RNA integrity was determined using an Agilent 2100 Bioanalyzer system and the RNA was stored at –80 °C until further analysis.

## RNA sequencing and analysis

The RNA sequencing library was prepared using an Illumina Stranded Total RNA Prep with Ribo-Zero Plus kit (San Diego, CA, USA) and sequenced using a NovaSeq 6000 SP Reagent Kit v1.5 in $2 \times 75$ bp configuration on an Illumina NovaSeq 6000 system (San Diego, CA, USA) at the LuxGen Platform (Dudelange, Luxembourg). Raw data files were cleaned using kneaddata (https://github.com/biobakery/kneaddata), including adapter removal, discarding reads less than half the expected length, and removal of reads mapping to contaminant databases (ribosomal RNA and mouse genome, in the case where host transcripts were not the target). Colonic tissue transcripts were mapped to the *Mus musculus* genome, while cecal content transcripts were mapped individually to each of the 14SM strain genomes (Table EV1) using Salmon (Patro et al, 2017). Host transcripts that were not found at least once on average across all samples were filtered out; for metatranscriptomic analyses, a cutoff of 2.5 cpm in at least 50% of the sample libraries was used, as described by Chen et al (Chen et al, 2016b). Differential expression analysis was carried out using DESeq2 1.30.1 with correction for multiple comparisons using the Benjamini–Hochberg method (Love et al, 2014). For metatranscriptomic analysis in HUMANn3, the forward and reverse fastq files were concatenated and then analysed by aligning to a custom taxonomic profile of the 14SM members and *C. rodentium*, followed by remapping of UniRef90 identifiers to Pfam identifiers (Beghini et al, 2021).

## Bacterial relative abundance quantification

Initial colonization of the SM was verified using phylotype-specific qPCR primers (Steimle et al, 2021). Analysis of the final relative abundances was performed following 16S rRNA gene sequencing of the V4 region using primers described by Kozich et al, (Kozich et al, 2013). Amplicon libraries were prepared using the Quick-16S™ NGS Library Prep Kit (BaseClear, Leiden, NL) and run on an Illumina MiSeq (San Diego, CA, USA) using the MiSeq Reagent Kit v2 (500-cycles) at the Integrated BioBank of Luxembourg (IBBL, Dudelange, Luxembourg). Raw, demultiplexed sequences were processed and analyzed using mothur version 1.41.3 (Schloss et al, 2009). Reads were assigned taxonomy using the k-Nearest Neighbor method based on a custom database containing the sequences of the 14SM bacteria and *C. rodentium*. Co-abundance correlation analyses were performed on relative abundances

(Dataset EV1) for 14SM FR, 14SM FF, 13SM FR, and 13SM FF mice using the corr.test() function of psych v2.4.1, with visualizations generated using the corrr package v0.4.4. (EV Fig. 4).

### *C. rodentium* culturing and enumeration

*C. rodentium* was cultivated aerobically in LB medium at 37 °C while shaking and CFUs were enumerated using LB-agar plates containing kanamycin, as previously described (Desai et al, 2016).

### Intestinal short-chain fatty acid analysis

Thirty to 100 mg of flash-frozen cecal content collected from uninfected mice was used for the fatty acid analysis by gas chromatography-mass spectrometry (GC-MS), as previously described (Greenhalgh et al, 2019; Wolter et al, 2021b).

### Immune cell profiling of colonic lamina propria

Colons from infected mice were opened longitudinally and gently washed with ice-cold 1X PBS before being placed in Hanks' buffered saline solution (HBSS) with phenol red and without calcium and magnesium (Lonza, Basel, Switzerland) on ice. Once all tissues were collected, immune cells from the colonic lamina propria were extracted using the Lamina Propria Dissociation Kit for mice and gentleMACS dissociator (Miltenyi Biotec, Bergisch Gladbach, Germany) following the manufacturer's instructions. Approximately $1.5 \times 10^6$ cells per animal were stained and acquired using a NovoCyte Quanteon flow cytometer (ACEA Biosciences Inc., San Diego, CA, USA), as previously described (Grant et al, 2023). The gating strategy can be found in Fig. EV3B.

### Statistical analyses

Unless otherwise stated, statistical analyses were performed using Prism 10.1.2 (GraphPad Software, Inc., San Diego, CA, USA). Outlier removal was performed using the ROUT method with a coefficient Q = 1% (Motulsky and Brown, 2006). Data were assessed for normality using the Kolmogorov–Smirnov test. Unless otherwise stated, two-way ANOVA followed by multiple pairwise comparisons (between diets within the same SM and between SM of the same diet) with $p$ values adjustment using the Benjamini–Hochberg method. For ordinal data, a two-way ordinal regression test was performed in R, using the package ordinal v2023.12-4 followed by pairwise multiple comparisons and $p$ values adjustment with the Benjamini–Hochberg method using emmeans v1.8.8. The specific test and the number of animals used for each experiment is detailed in the figure legends.

## Data availability

The datasets produced in this study are available in the following databases: • 16S rDNA and RNA-Seq data: European Nucleotide Archive (ENA) at EMBL-EBI PRJEB51371. • Flow cytometry data: FlowRepository FR-FCM-Z7A7 (http://flowrepository.org).

The source data of this paper are collected in the following database record: biostudies:S-SCDT-10_1038-S44320-024-00036-7.

## Peer review information

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

## Acknowledgements

We are extremely grateful for support from the Luxembourg National Research Fund (FNR) for CORE grants (C15/BM/10318186 and C18/BM/12585940) and BRIDGES grant (22/17426243) to MSD. We thank the US National Institutes of Health for funding ECM (DK118024 and DK125445). MW was supported by a Fulbright grant for Visiting Scholars from the Commission for Educational Exchange between the United States of America, Belgium, and Luxembourg. ETG was supported by FNR PRIDE (17/11823097) and the Fondation du Pélican de Mie et Pierre Hippert-Faber, under the aegis of the Fondation de Luxembourg. MB was supported by a European Commission Horizon 2020 Marie Skłodowska-Curie Actions individual fellowship (897408). We thank MEDICE Arzneimittel Pütter GmbH & Co. KG, Germany and Theralution GmbH, Germany, for funding through the public–private partnership FNR BRIDGES grant (22/17426243). We also thank Gunnar Hansson and George Birchenough (University of Gothenburg, Sweden) for helping us set up the assay for measuring mucus penetrability. Finally, we thank the University of Michigan Germfree Core and Microbiome Core for expert technical assistance; Sophie Craig for her technical contributions in the lab of MSD; Nathalie Nicot at the LuxGen Platform of the Luxembourg Institute of Health and the Laboratoire national de santé for her support in the RNA library preparation and sequencing; Brian De Witt, Lorieza Castillo, and Wim Ammerlaan at the Integrated BioBank of Luxembourg (IBBL) for V4 16S rRNA gene sequencing assistance; and Christian Jäger, Xiangyi Dong, and Floriane Gavotto from the Luxembourg Centre for Systems Biomedicine Metabolomics Platform for their support with the SCFA analysis. We also thank Markus Ollert for their encouragement and advice. For the purpose of open access, and in fulfillment of the obligations arising from the grant agreement, the author has applied a Creative Commons Attribution 4.0 International (CC BY 4.0) license to any Author Accepted Manuscript version arising from this submission.

## Author contributions

**Mathis Wolter**: Conceptualization; Data curation; Formal analysis; Validation; Investigation; Visualization; Methodology; Writing—original draft; Writing—review and editing. **Erica T Grant**: Data curation; Formal analysis; Investigation; Visualization; Methodology; Writing—review and editing. **Marie Boudaud**: Data curation; Formal analysis; Investigation; Visualization; Methodology; Writing—review and editing. **Nicholas A Pudlo**: Investigation; Methodology; Writing—review and editing. **Gabriel V Pereira**: Investigation; Writing—review and editing. **Kathryn A Eaton**: Investigation; Methodology; Writing—review and editing. **Eric C Martens**: Conceptualization; Resources; Supervision; Methodology; Writing—review and editing. **Mahesh S Desai**: Conceptualization; Resources; Supervision; Funding acquisition; Investigation; Visualization; Methodology; Writing—original draft; Project administration; Writing—review and editing.

Source data underlying figure panels in this paper may have individual authorship assigned. Where available, figure panel/source data authorship is

listed in the following database record: biostudies:S-SCDT-10_1038-S44320-024-00036-7.

## Disclosure and competing interests statement

Mahesh S Desai works as a consultant and an advisory board member at Theralution GmbH, Germany. Eric C. Martens works as a consultant and an advisory board member at January, Inc., United States.

# Expanded View Figures

**Figure EV1.   Synthetic microbiota profiles through colonization, feeding period, and *Citrobacter rodentium* infection.**

Relative abundances of 14SM, 13SM, 12SM, 11SM, and 10SM bacteria among mice fed a fiber-rich (FR, left panels) or fiber-free (FF, right panels) diet. Abundances were calculated from 16S rRNA gene sequencing results after initial colonization (day 6–8, $n = 0$–9), prior to diet change (day 10–13, $n = 4$–13), during the 40-day feeding period (day 15–53, $n = 3$–14), or during *Citrobacter rodentium* infection (day 54–64, $n = 4$–13). The number of sampling points varies by SM, with 31–39 for 14SM, 4 for 13SM, 33–34 for 12SM, 15–16 for 11SM, and 16–18 for 10SM.

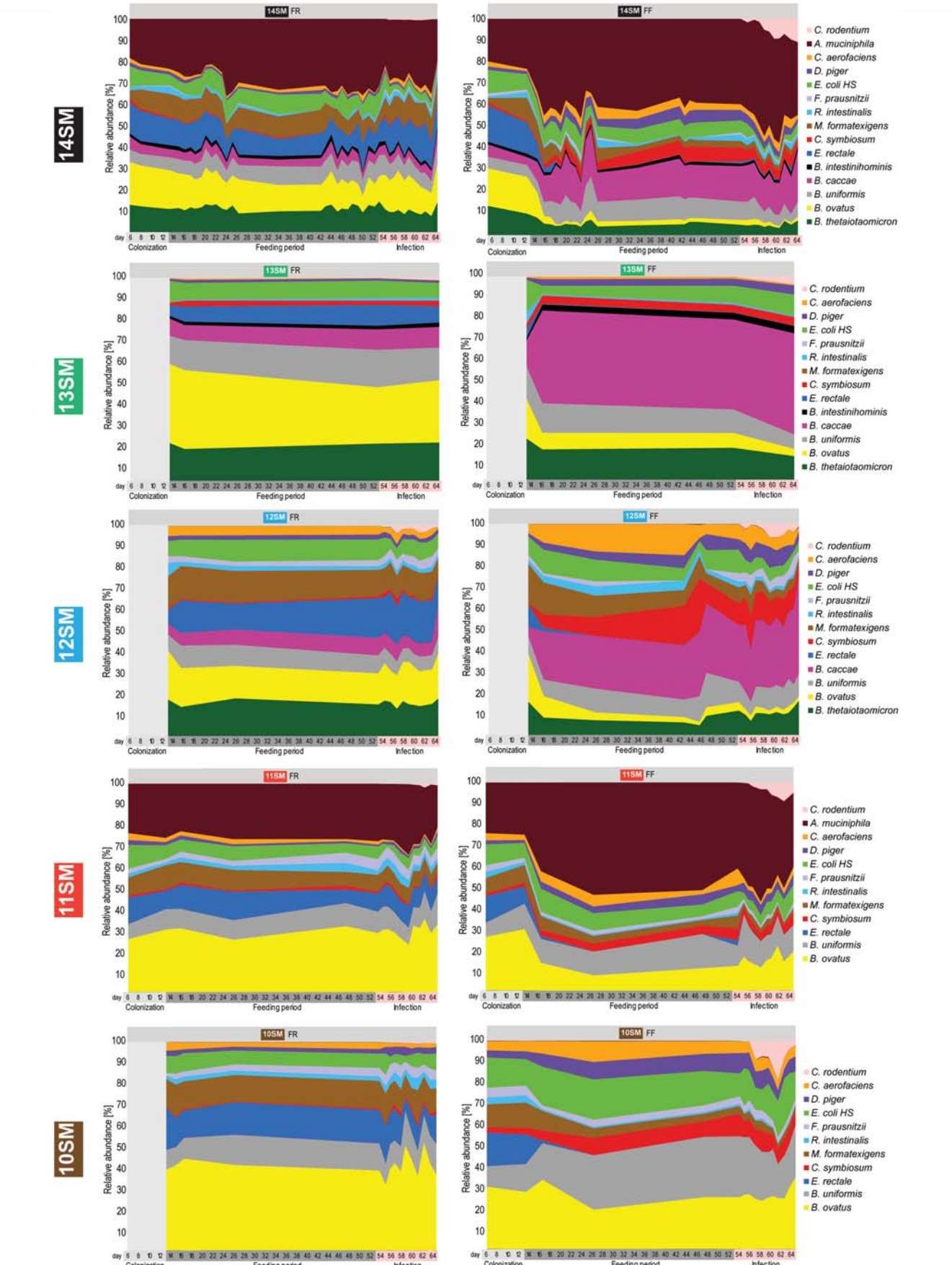

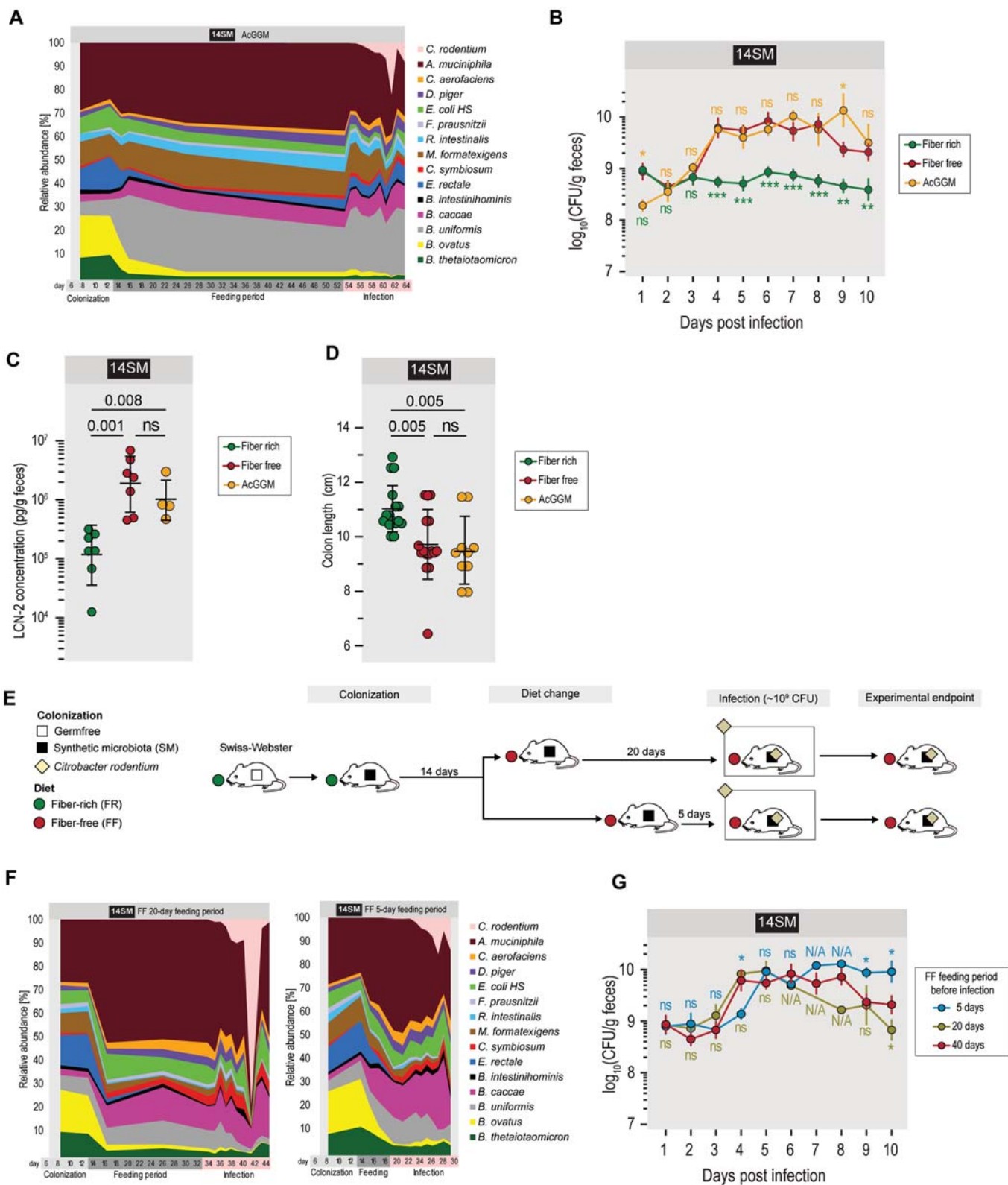

◀ **Figure EV2. Acetylated galactoglucomannan supplementation is insufficient to rescue the resistant phenotype of the FR-fed mice, while short-term feeding of the FF diet is sufficient to develop a susceptible phenotype.**

(**A**) Relative abundances of 14SM bacteria among mice fed a fiber-free (FF) diet supplemented with 7.5% acetylated galactoglucomannan (AcGGM). Abundances were calculated from 16S rRNA gene sequencing results after initial colonization (day 8), prior to diet change (day 13), during the feeding period (day 15–54), or during *C. rodentium* infection. Ten sampling points, $n = 1$–5 per sampling point. (**B**) Fecal *C. rodentium* load of 14SM FR-, FF- and AcGGM-fed Swiss Webster mice during the 10 days of infection ($n = 5$–11 per group following the exclusion of three outliers). Error bars show SEM and the curves the arithmetic mean; one-way ANOVA with adjustment for multiple comparisons between FF and AcGGM (top significance labels, yellow text) or between FF and FR (bottom significance labels, green text) group using the Benjamini–Hochberg method. ns, non-significant; *, adjusted $p < 0.1$; **, adjusted $p < 0.01$; ***, adjusted $p < 0.001$. (**C**) Fecal LCN-2 concentration of infected 14SM FR-, FF-, and AcGGM-fed mice on the final day of their infection ($n = 4$–7 per group, no outliers). Error bars show SD and the center of the arithmetic mean; one-way ANOVA with adjustment for multiple comparisons using the Benjamini–Hochberg method. ns non-significant. (**D**) Colon length of infected 14SM FR-, FF-, and AcGGM-fed mice on the final day of their infection ($n = 1$–16 per group, no outliers). Error bars show SD and the center of the arithmetic mean; one-way ANOVA with adjustment for multiple comparisons using the Benjamini–Hochberg method. ns non-significant. (**E**) Experimental timeline for short-term feeding experiment. Age-matched germ-free Swiss Webster mice were gavaged with one of the different synthetic gut microbiota (SM) on 3 consecutive days. These mice were maintained for at least 14 days on the fiber-rich (FR) diet, after which mice were switched to a fiber-free (FF) for either 20 days (FF, $n = 3$) or 5 days (FF, $n = 4$) before they were infected with *Citrobacter rodentium*. After infection, mice were closely observed for up to 10 days. (**F**) Relative abundances of 14SM bacteria among mice fed a fiber-free diet for 20 (left) and 5 (right) days prior to infection. Abundances were calculated from 16S rRNA gene sequencing results, and the number of sampling points varies by feeding regimen, with 16 for the FF diet with a 20-day feeding period and 10 for the FF diet with a 5-day feeding period, $n = 1$–4 mice per sampling point. (**G**) Fecal *C. rodentium* load of 14SM mice fed the FF diet for 5, 20, or 40 days before infection. 5-day feeding $n = 4$, 20-day feeding $n = 3$, 40-day feeding $n = 12$. Some days it was impossible to recover fecal pellets from certain mice. Error bars show SEM and the curves the arithmetic mean; one-way ANOVA with adjustment for multiple comparisons between 5-day feeding and 40-day feeding (top significance labels, blue text) groups or between 20-day feeding and 40-day feeding (bottom significance labels, olive green text) groups using the Benjamini–Hochberg method. N/A, no statistic available as $n = 1$ for one group; ns non-significant; *, adjusted $p < 0.1$; **, adjusted $p < 0.01$; ***, adjusted $p < 0.001$.

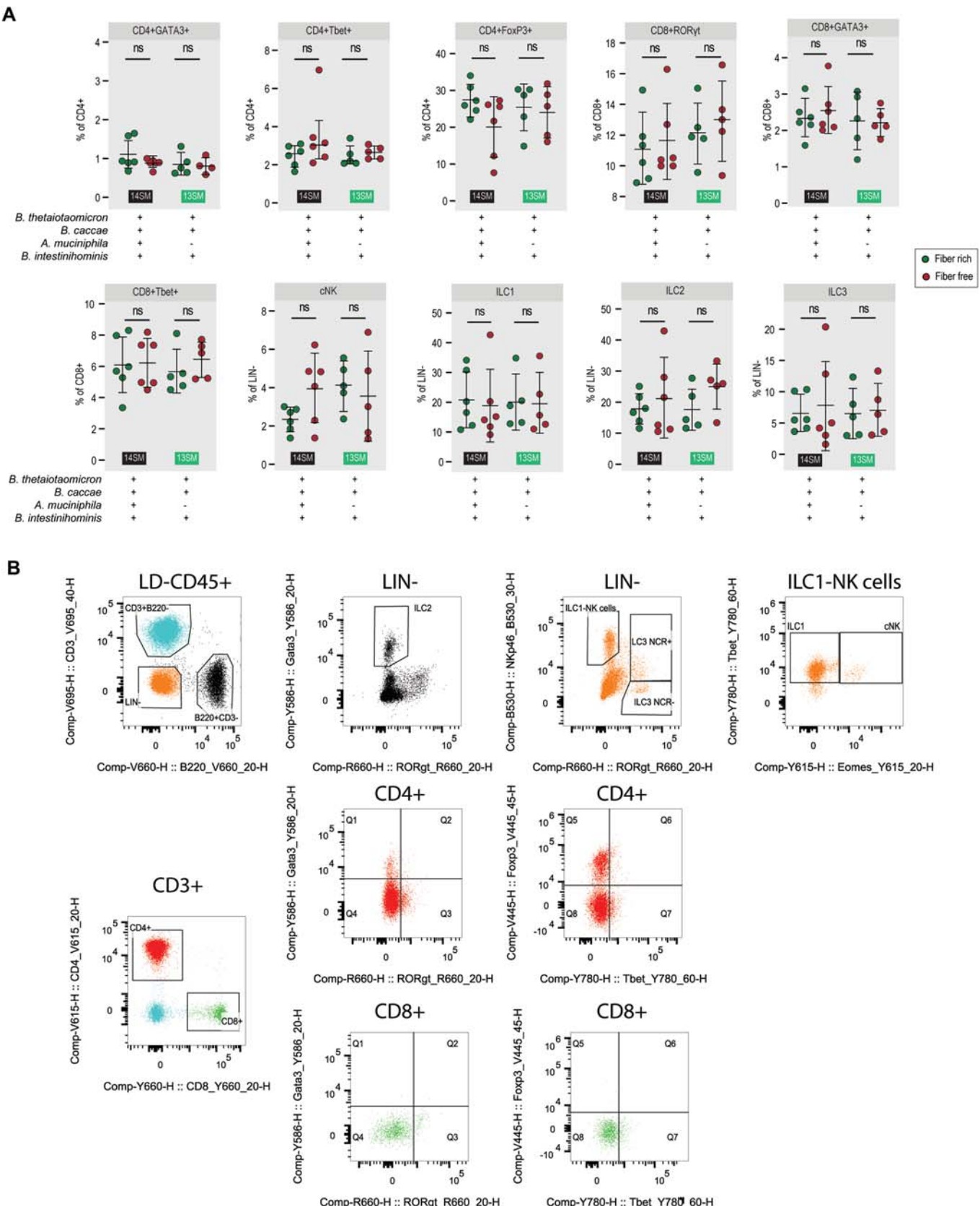

**Figure EV3.   Host immune cell populations remain largely unchanged between dietary groups after infection with C. rodentium.**

(A) Non-significantly altered immune cell population as a percent of the parent population among mice at 3 DPI, as determined by fluorescence-activated cell sorting (FACS). Population percentages were analyzed by two-way ANOVA (main effects only) with adjustment for multiple comparisons between SMs of the same diet (shown) and between diets of the same SM (see Source Data) using the Benjamini–Hochberg method ($n = 4$–6 per group, following exclusion of one outlier in 13SM FF CD + GATA3+ group). Error bars show SD and the center of the arithmetic mean; ns non-significant (adjusted $p > 0.1$). (B) Gating strategy for FACS analysis.

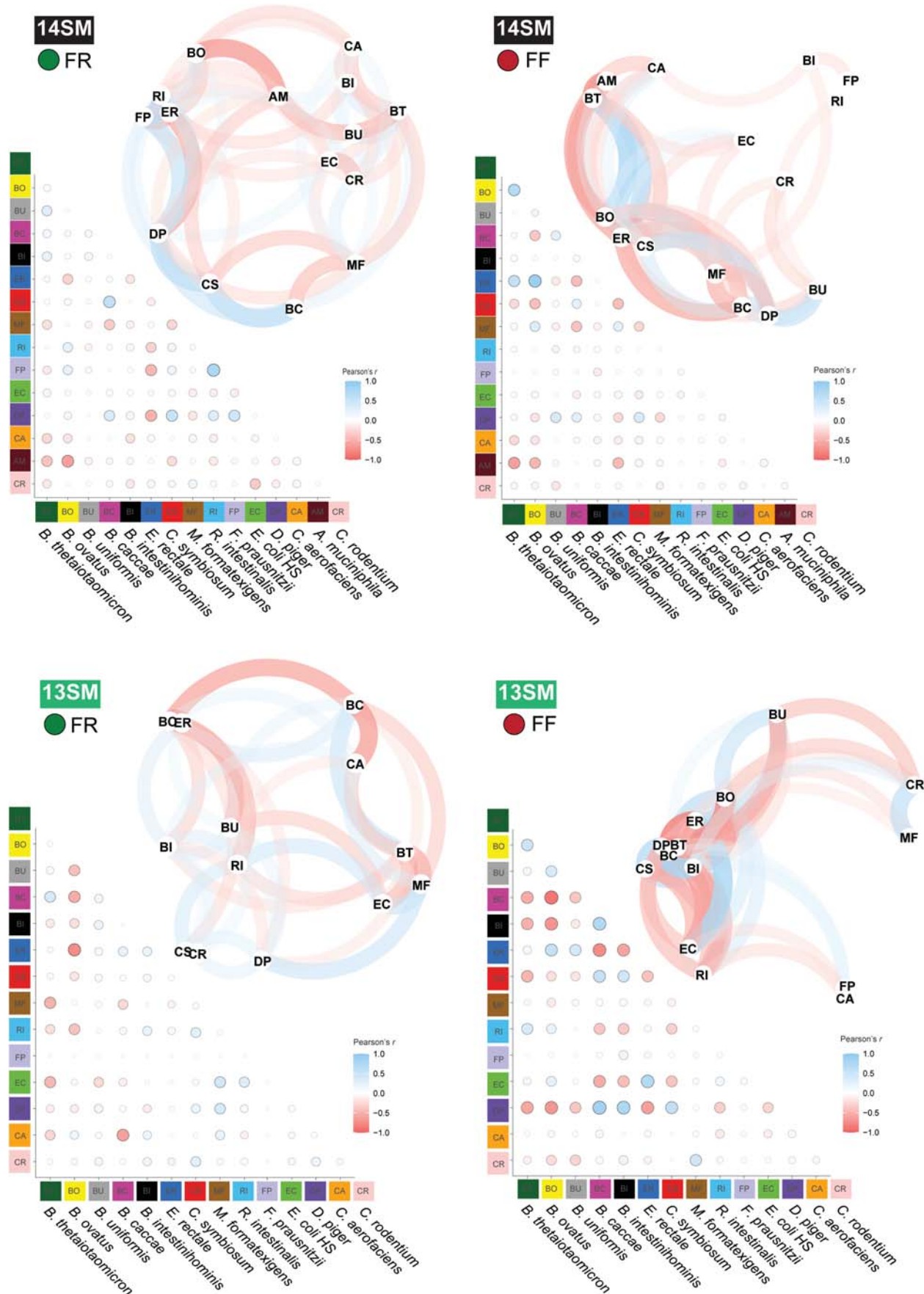

**Figure EV4. Distinct effects of diet and community composition on bacterial co-abundance correlations.**

Correlation matrices and networks of relative abundances in Dataset EV1 across all timepoints (14 FR, $n = 177$; 14SM FF, $n = 200$; 13SM FR, $n = 20$; 13SM FF, $n = 19$). In the matrix (lower left), all correlations are shown. Correlations were calculated using the Pearson method. In the network plot (upper right), only correlations with $|r| > 0.2$ are shown.

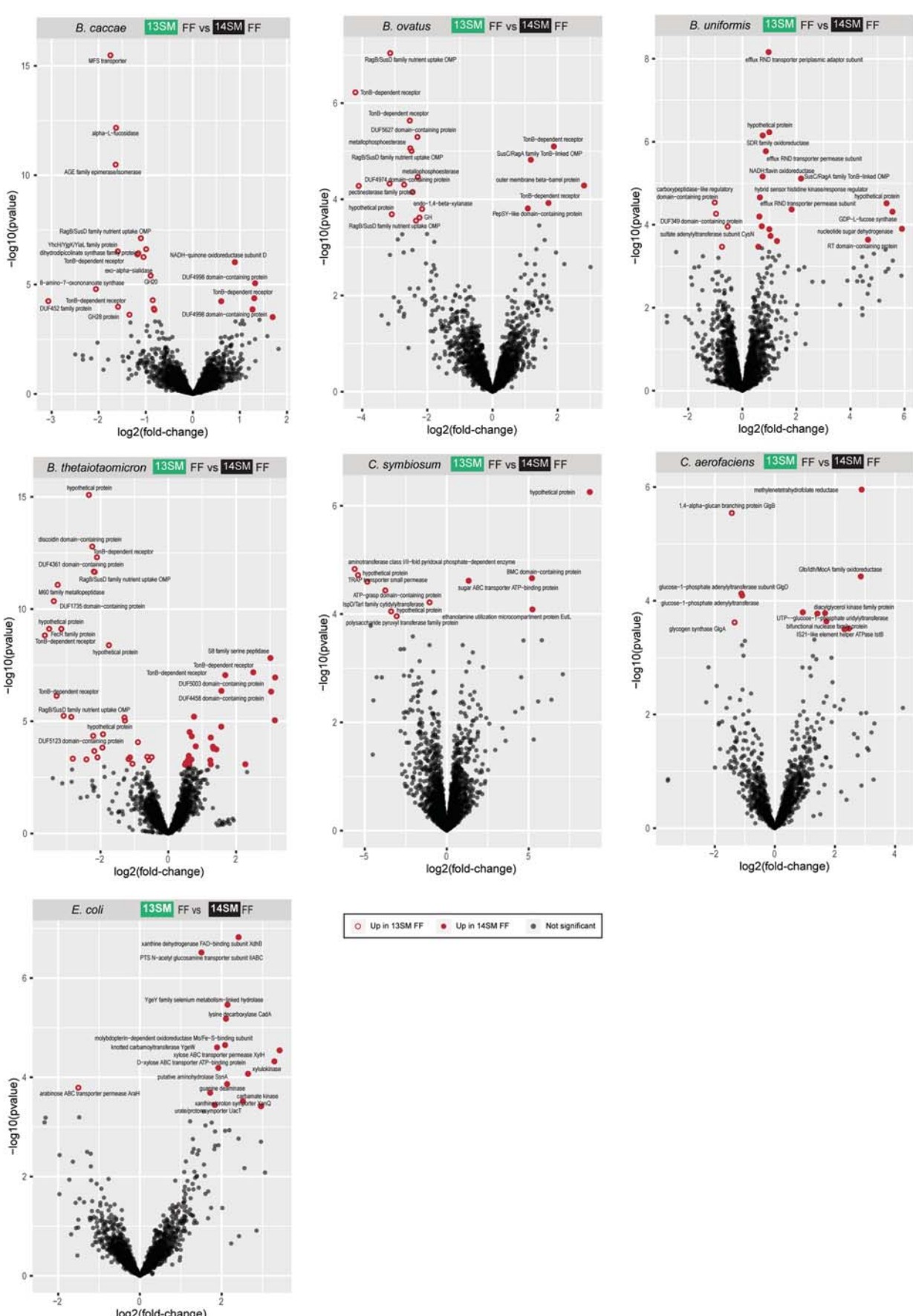

◀

**Figure EV5. Bacterial transcriptomes altered by *A. muciniphila* presence in fiber-deprived mice.**

Volcano plots of synthetic microbiota members with >10 DEGs in the 13SM FF vs 14SM FF contrast (*B. caccae, B. ovatus, B. thetaiotaomicron, B. uniformis, C. symbiosum, C. aerofaciens,* and *E. coli*). RNA was extracted from cecal contents at 3 DPI in 13SM FF (*C. rodentium* resistant) and 14SM FF (*C. rodentium* susceptible) mice and individually mapped to the genome specified in Salmon. Features were filtered out if <2.5 cpm in four samples. Significance based on adjusted *p* value <0.05 based on Wald test adjusted using the Benjamini–Hochberg method in DESeq2. GH glycoside hydrolase, OMP outer membrane protein, RT reverse transcriptase.

