## [Peer Review File · Molecular Systems Biology]

Diet-driven differential response of *Akkermansia muciniphila* modulates pathogen susceptibility

Mahesh Desai, Mathis Wolter, Erica Grant, Marie Boudaud, Nicholas Pudlo, Gabriel Pereira, Kathryn Eaton, and Eric Martens

Corresponding author(s): Mahesh Desai (mahesh.desai@lih.lu)

Review Timeline:

Submission Date:	7th Dec 23
Editorial Decision:	1st Feb 24
Revision Received:	27th Mar 24
Editorial Decision:	3rd Apr 24
Revision Received:	6th Apr 24
Accepted:	12th Apr 24

Editor: Maria Polychronidou

Transaction Report:

1st Feb 2024

Manuscript Number: MSB-2023-12163

Title: Diet-driven differential response of Akkermansia muciniphila modulates pathogen susceptibility

Dear Mahesh,

Thank you again for submitting your work to Molecular Systems Biology. We have now heard back from two of the three reviewers who agreed to evaluate your study. In the interest of time and given that the recommendations of the two reviewers are similar we have decided to proceed with making a decision based on these two available reports. Overall, the reviewers are supportive and feel that the study is a relevant contribution to the field. However, they do raise a series of concerns, which we would invite you to address in a revision.

I think that the recommendations of the reviewers seem clear and I therefore do not see the need to repeat any of their comments here. All issues raised need to be satisfactorily addressed. As you may already know, our editorial policy allows in principle a single round of major revision, so it is essential to provide responses to the reviewers' comments that are as complete as possible. Please feel free to contact me in case you would like to discuss in further detail any of the issues raised or if you would like to share your revision plan with me. I would be happy to schedule a call.

On a more editorial level, we would ask you to address the following points:

- Appendix Table 1 should be provided as Table EV1. It can be provided as a doc or xls.
- Please provide a "standfirst text" summarizing the study in one or two sentences (approximately 250 characters), three to four "bullet points" highlighting the main findings and a "synopsis image" (550px width and max 400px height, jpeg format) to highlight the paper on our homepage.
- All Materials and Methods need to be described in the main text. We would encourage you to use 'Structured Methods', our new Materials and Methods format. According to this format, the Materials and Methods section should include a Reagents and Tools Table (listing key reagents, experimental models, software and relevant equipment and including their sources and relevant identifiers) followed by a Methods and Protocols section in which we encourage the authors to describe their methods using a step-by-step protocol format with bullet points, to facilitate the adoption of the methodologies across labs. More information on how to adhere to this format as well as downloadable templates (.doc or .xls) for the Reagents and Tools Table can be found in our author guidelines: . An example of a Method paper with Structured Methods can be found here:
- Please include a Data availability section describing how the data, code etc. have been made available. This section needs to be formatted according to the example below:
The datasets and computer code produced in this study are available in the following databases:
 - Chip-Seq data: Gene Expression Omnibus GSE46748 (<https://www.ncbi.nlm.nih.gov/geo/query/acc.cgi?acc=GSE46748>)
 - Modeling computer scripts: GitHub (<https://github.com/SysBioChalmers/GECKO/releases/tag/v1.0>)
 - [data type]: [full name of the resource] [accession number/identifier] ([doi or URL or identifiers.org/DATABASE:ACCESSION])
- For data quantification: please specify the name of the statistical test used to generate error bars and P values, the number (n) of independent experiments (specify technical or biological replicates) underlying each data point and the test used to calculate p-values in each figure legend. The figure legends should contain a basic description of n, P and the test applied. Graphs must include a description of the bars and the error bars (s.d., s.e.m.).
- When you resubmit your manuscript, please download our CHECKLIST (<https://bit.ly/EMBOPressAuthorChecklist>) and include the completed form in your submission.
Please note that the Author Checklist will be published alongside the paper as part of the transparent process (<https://www.embopress.org/page/journal/17444292/authorguide#transparentprocess>).

If you feel you can satisfactorily deal with these points and those listed by the referees, you may wish to submit a revised version of your manuscript. Please attach a covering letter giving details of the way in which you have handled each of the points raised by the referees. A revised manuscript will be once again subject to review and you probably understand that we can give you no guarantee at this stage that the eventual outcome will be favorable.

Kind regards,

Maria

We realize that it is difficult to revise to a specific deadline. In the interest of protecting the conceptual advance provided by the work, we recommend a revision within 3 months (1st May 2024). Please discuss the revision progress ahead of this time with the editor if you require more time to complete the revisions. Use the link below to submit your revision:

IMPORTANT: When you send your revision, we will require the following items:

1. the manuscript text in LaTeX, RTF or MS Word format
2. a letter with a detailed description of the changes made in response to the referees. Please specify clearly the exact places in the text (pages and paragraphs) where each change has been made in response to each specific comment given
3. three to four 'bullet points' highlighting the main findings of your study
4. a short 'blurb' text summarizing in two sentences the study (max. 250 characters)
5. a 'thumbnail image' (550px width and max 400px height, Illustrator, PowerPoint or jpeg format), which can be used as 'visual title' for the synopsis section of your paper.

6. Please include an author contributions statement after the Acknowledgements section (see

<https://www.embopress.org/page/journal/17444292/authorguide>)

7. Please complete the CHECKLIST available at (<https://bit.ly/EMBOPressAuthorChecklist>).

Please note that the Author Checklist will be published alongside the paper as part of the transparent process

(<https://www.embopress.org/page/journal/17444292/authorguide#transparentprocess>).

See also figure legend guidelines: <https://www.embopress.org/page/journal/17444292/authorguide#figureformat>

9. Please note that corresponding authors are required to supply an ORCID ID for their name upon submission of a revised manuscript (EMBO Press signed a joint statement to encourage ORCID adoption).

(<https://www.embopress.org/page/journal/17444292/authorguide#editorialprocess>)

Currently, our records indicate that the ORCID for your account is 0000-0002-9223-2209.

Link Not Available

*** PLEASE NOTE *** As part of the EMBO Press transparent editorial process initiative (see our Editorial at <https://dx.doi.org/10.1038/msb.2010.72>), Molecular Systems Biology publishes online a Review Process File with each accepted manuscripts. This file will be published in conjunction with your paper and will include the anonymous referee reports, your point-by-point response and all pertinent correspondence relating to the manuscript. If you do NOT want this File to be published, please inform the editorial office at msb@embo.org within 14 days upon receipt of the present letter.

Reviewer #1:

Wolter, Desai and colleagues present a novel and carefully performed study examining the effects of Akkermansia on aspects of colonic biology. The most interesting finding is that it seems to contribute to the detrimental effects of a fiber-free diet in a *Citrobacter rodentium* model. While a smoking-gun mechanism is not clearly identified, it is not due to a lack of effort, as they comprehensively profile many aspects of microbiome and host biology. The methodologies employed are state-of-the-art and very carefully performed. I have a few critiques that I think would strengthen what is already a very mature manuscript.

- The authors cite a relevant manuscript by Liang et al, where sialic acid was found to be important for supporting *C. rodentium* infection. It seems reasonable that *A. muciniphila* could be increasing the availability of sialic acid - can the authors quantify

sialic acid and other monosaccharide-like compounds that *C. rodentium* can utilize as carbon sources?

- Could *A. muciniphila* be mediating its supportive effects on *C. rodentium* via other members of the microbiome? Quantifying pair-wise correlations between all bacterial strains and examining for strains that are positively correlated with *C. rodentium* or *A. muciniphila* could potentially be instructive.

- The authors should be careful with the interpretation of their results. In Figs 1 and 2, they focus their analyses on FR vs FF. This is fine, but they then tend to indirectly draw conclusions comparing FF-14SM vs FF-13SM or FF-11SM vs FF-10SM without formally testing for differences. Stated another way, in a strain consortium where there is loss of a difference between FR vs FF, they tend to attribute that to a change in the microbiota composition. Later in the manuscript, they do formally compare FF 14SM vs FF 13SM, but they do not do these comparisons in Figs 1-2 and they should. The authors may be right - FF 14SM vs FF 13SM may indeed be different for the readouts in Figs 1-2, but loss of a difference in the FF setting is not the same thing and could arise be due to too much heterogeneity or inadequate statistical power.

- A search for prior manuscripts linking *A. muciniphila* and *C. rodentium* identifies this manuscript:

<https://pubmed.ncbi.nlm.nih.gov/34592891/>

where apparently these authors thought *Akkermansia* protected against *C. rodentium*-induced colitis. Perhaps this is consistent with the observation in the current manuscript that *A. muciniphila* as the sole mucin-degrader appears to be protective to *C. rodentium* colonization in the context of a fiber-sufficient diet?

Reviewer #2:

The report by Wolter and colleagues addresses the role of mucolytic bacteria in the GI tract and pathogen susceptibility. By using defined microbial communities and diets, the authors elegantly outline the complexity and context-dependency (diet) of the impact of mucus degradation. Prior models wherein mucin degradation was viewed as inherently pathogenic appears to be more nuanced and that not all mucolytic bacteria, even specialists, have the same impact and that not all fiber has same consequences on host resistance. The role of *Akkermansia* is specifically clarified in that under fiber replete conditions, *Akkermansia* is mildly protective to *Citrobacter*-induced colitis, but can be pathogenic in the context of fiber-depleted diets. This phenotype is not merely the result of a decrease mucus thickness as communities lacking *Akkermansia* still caused thinning of the mucus layer. The molecular mechanisms that underlies this susceptibility is still unclear with no clear immunophenotype or transcriptional signatures but correlates with increased mucus penetrability, although this phenotype is milder (Fig 4b). The manuscript is well written and organized. The experiments are well controlled and the results of high quality. There is no major "punchline" but the results highlight the complex microbe-microbe interaction, even in these simplified communities, and that *Akkermansia* interactions with their host are also more complex than simply their ability to consume mucin. There are also useful cautionary tales regarding extrapolation from behavior of bacteria grown in monoculture.

MSB-2023-12163 Response to Reviewers

Reviewer #1:

Wolter, Desai and colleagues present a novel and carefully performed study examining the effects of Akkermansia on aspects of colonic biology. The most interesting finding is that it seems to contribute to the detrimental effects of a fiber-free diet in a Citrobacter rodentium model. While a smoking-gun mechanism is not clearly identified, it is not due to a lack of effort, as they comprehensively profile many aspects of microbiome and host biology. The methodologies employed are state-of-the-art and very carefully performed. I have a few critiques that I think would strengthen what is already a very mature manuscript.

Thank you for this concise summary of our submitted work.

- The authors cite a relevant manuscript by Liang et al, where sialic acid was found to be important for supporting *C. rodentium* infection. It seems reasonable that *A. muciniphila* could be increasing the availability of sialic acid - can the authors quantify sialic acid and other monosaccharide-like compounds that *C. rodentium* can utilize as carbon sources? Indeed it would be interesting to compare our results with that of Liang et al either by measuring free sialic acid in the stool, using a commercially available kit, or even using an adaptation of a previously published protocol to quantify enzyme activities (DOI: 10.1016/j.xpro.2021.100326). Unfortunately, we do not have any remaining stool samples from these mice, so we would have to repeat the mouse work in order to address these points. Given the revision timeline of three months, additional mouse work is not feasible. Nonetheless, in Figure 4d, we do observe increased sialidase transcription on the FF diet, which would suggest that there may be greater availability of sialic acid to be used by *C. rodentium*. Moreover, since *A. muciniphila* is known to produce sialidases to degrade mucin but does not consume sialic acid (DOI: 10.1038/s41467-023-37533-6), it is indeed reasonable to predict that increased mucin degradation by *A. muciniphila* might promote *C. rodentium* growth through this metabolite. We have included this observation in the discussion where we previously highlighted the Liang et al. paper to encourage the future investigation of this aspect in *A. muciniphila*-containing communities (lines 411-421):

Changes in which mucosal sugars are released in the presence or absence of A. muciniphila under fiber-deprived conditions could also play a role in the susceptibility, as it was recently shown that sialic acid—a sugar present in mucin glycoproteins—plays a vital role in aiding transition of C. rodentium from lumen to mucosa (Liang et al, 2023). Although we did not measure free sialic acid directly, we detect an increase in sialidase transcripts among mice fed an FF diet (Fig. 4d), suggesting that more sialic acid might be available as a result of altered microbial activities under fiber-deprived conditions. Furthermore, A. muciniphila can release sialic acid during the degradation of mucin, yet its inability to subsist on sialic acid alone suggests that A. muciniphila does not use this compound for growth (Shuoker et al, 2023). It is therefore plausible that the combination of a FF diet and presence of A. muciniphila results in the highest sialic acid availability and thereby facilitates the invasion of C. rodentium, which should be verified in future experiments.”

- Could *A. muciniphila* be mediating its supportive effects on *C. rodentium* via other members of the microbiome? Quantifying pair-wise correlations between all bacterial strains and examining for strains that are positively correlated with *C. rodentium* or *A. muciniphila* could potentially be instructive.

We appreciate the reviewer's suggestion and agree this is a worthwhile possibility to explore. We therefore implemented correlation analyses on relative abundances from Dataset EV1 plot for 14SM FR, 14SM FF, 13SM FR, and 13SM FF mice using the corr.test() function of psych v2.4.1, with visualizations generated using the corrr package v0.4.4. The correlation matrix and network plot have been included in a new Fig EV4 (former ED Fig 4 shifted to ED Fig 5) and we provide a brief summary of this analysis in the Results section (lines 285-295).

*“In order to assess whether the FF diet and the presence of *A. muciniphila* might be indirectly linked to *C. rodentium* susceptibility—i.e. through its influence on other community members—we performed pairwise correlation analyses on the abundances of each bacterium across 13SM and 14SM mice fed an FR or FF diet (Fig. EV4). Although we identified a number of statistically significant bacterial abundance co-associations that were altered between the two diets or between 13SM and 14SM colonized mice, we were unable to identify positive associations shared by both *A. muciniphila* and *C. rodentium* that were exclusively present in 14SM FF-fed mice and might thereby explain the increased susceptibility to infection. However, there was a significant negative correlation between *Marvinbryantia formatexigens* and both *A. muciniphila* ($r = -0.171$) and *C. rodentium* ($r = -0.232$), which was only observed within 14SM FF mice, suggesting that the acetogen *M. formatexigens* could be conferring some protection against infection under the FR and 13SM conditions.”*

- The authors should be careful with the interpretation of their results. In Figs 1 and 2, they focus their analyses on FR vs FF. This is fine, but they then tend to indirectly draw conclusions comparing FF-14SM vs FF-13SM or FF-11SM vs FF-10SM without formally testing for differences. Stated another way, in a strain consortium where there is loss of a difference between FR vs FF, they tend to attribute that to a change in the microbiota composition. Later in the manuscript, they do formally compare FF 14SM vs FF 13SM, but they do not do these comparisons in Figs 1-2 and they should. The authors may be right - FF 14SM vs FF 13SM may indeed be different for the readouts in Figs 1-2, but loss of a difference in the FF setting is not the same thing and could arise be due to too much heterogeneity or inadequate statistical power.

The reviewer’s point regarding formal testing for inter-SM comparisons is well taken and we have modified the statistical analyses accordingly and included the full test results (i.e. with multiple comparisons between SMs of same diet and between diets of same SM) in the Source Data for the figure, which is linked in the legend. As a consequence of this change, the adjusted p-values shown in the figures have changed slightly (not altering main conclusions). Nonetheless, we have elected not to include all significantly different groups on the plots in order to maintain readability and focus attention on the key comparisons. Of course, if the reviewer strongly objects to this solution and prefers that we also include the p-values for these between SM comparisons in Fig 1 and 2, we can possibly integrate this information in the figure, however this would make the already full-sized figures appear cluttered.

- A search for prior manuscripts linking *A. muciniphila* and *C. rodentium* identifies this manuscript:

<https://pubmed.ncbi.nlm.nih.gov/34592891/>

where apparently these authors thought *Akkermansia* protected against *C. rodentium*-induced colitis. Perhaps this is consistent with the observation in the current manuscript that *A. muciniphila* as the sole mucin-degrader appears to be protective to *C. rodentium* colonization in the context of a fiber-sufficient diet?

Indeed, we agree that the premise for the referenced article is in line with the apparent protective effect of AM under the FR diet. We have added a brief reference to this article to further support the discussion surrounding AM’s diet-dependent behavior (lines 372-378):

*As *A. muciniphila* is regarded as a potential probiotic bacterium (Zhai et al, 2019; Ashrafian et al, 2021; Cani et al, 2022; Daniel et al, 2023), the effects of this bacterium on the activity of other members of the gut microbial community and under different dietary regimens must be considered when designing probiotic therapies using this microbe. Indeed Mao et al. show that treating mice with glycosaminoglycan polymer hyaluronan on a normal chow impacts the abundance of various members of the microbiome, enriching *A. muciniphila* in particular, which in turn confers protection against *C. rodentium* infection (Mao et al, 2021).*

Reviewer #2:

The report by Wolter and colleagues addresses the role of mucolytic bacteria in the GI tract and pathogen susceptibility. By using defined microbial communities and diets, the authors elegantly outline the complexity and context-dependency (diet) of the impact of mucus degradation. Prior models wherein mucin degradation was viewed as inherently pathogenic appears to be more nuanced and that not all mucolytic bacteria, even specialists, have the same impact and that not all fiber has same consequences on host resistance. The role of Akkermansia is specifically clarified in that under fiber replete conditions, Akkermansia is mildly protective to Citrobacter-induced colitis, but can be pathogenic in the context of fiber-depleted diets. This phenotype is not merely the result of a decrease mucus thickness as communities lacking Akkermansia still caused thinning of the mucus layer. The molecular mechanisms that underlies this susceptibility is still unclear with no clear immunophenotype or transcriptional signatures but correlates with increased mucus penetrability, although this phenotype is miled (Fig 4b).

The manuscript is well written and organized. The experiments are well controlled and the results of high quality. There is no major "punchline" but the results highlight the complex microbe-microbe interaction, even in these simplified communities, and that Akkermansia interactions with their host are also more complex than simply their ability to consume mucin. There are also useful cautionary tales regarding extrapolation from behavior of bacteria grown in monoculture.

We appreciate the reviewer's summary and assessment of the manuscript, in particular regarding the nuanced nature of mucin-degrading bacteria. We do not identify a specific suggestion within this feedback and therefore have no edits to report.

Reviewer #3

Wolter et al. explore the contribution of different mucin-degrading bacteria to pathogen susceptibility in mice. The authors use a strain drop out approach and different diets to highlight the conditions where the pathogen is causing the most damage. The number of different parameters tested to get to their different conclusions is very impressive (colon shortening, LCN-2 levels, weight loss, pathogen load, gene expression profiles for both host and microbes, mucosal thickness, penetrability, etc etc). This study highlights the complexity between gut microbiota composition, diet and other factors influence inflammatory conditions of the large intestine. Furthermore, the results from this study highlight the need to refrain from drawing extrapolated conclusions from more mechanistic studies on gut microbes individually.

The manuscript is really well written, with explanations along the way for each result where required. The presentation of the Figures is also excellent.

- Great introductions - sets the premise of the study really well

Thank you for this comment.

- Line 70 - strain not stain?

Many thanks for catching this typo—it is indeed strain.

- Line 107 - why are these non-mucin degrading species thriving?

It is not completely clear how these bacteria benefit (that is, which precise metabolites they may be using for growth), but we expect that it is through cross-feeding processes since they lack the necessary enzymes to degrade mucin directly. We have therefore added this brief explanation and reference to a review which well encapsulates this concept (lines 107-112, also below):

Among the non-mucin-degrading bacteria, Desulfovibrio piger, Clostridium symbiosum and Collinsella aerofaciens also thrived under FF conditions (Fig 1e), which we expect may be a result of free amino acids or other liberated metabolites that are not utilized by the primary mucin-degraders (Fischbach & Sonnenburg, 2011) or in case of D. piger may be owing to the release of terminal sulfate groups of mucins (Rey et al., 2013).

• Line 180 - why do the authors think that *B. uniformis* expands under these conditions? La Rosa 2019 does see an increase in this species

We confirm that we observed an expansion of *B. uniformis* in the 14SM+AcGGM mice rapidly after switching the diet (see timeline at bottom of the streamplot in Fig. EV2a), as was also observed by La Rosa 2019 (DOI: 10.1038/s41467-019-08812-y). While *B. uniformis* is not as selective as *Roseburia intestinalis*, it does possess RagD, SusB starch utilization loci and beta-galactosidases and can therefore also subsist on this substrate. This capacity is demonstrated more clearly in another publication by La Rosa et al (see Fig 3a in DOI: 10.1128/msphere.00554-18). We have provided a potential explanation for this phenomenon in the text (lines 176-180), also below:

Although AcGGM is primarily degraded by Roseburia intestinalis, Faecalibacterium prausnitzii, and Eubacterium rectale (La Rosa et al, 2019), three butyrogenic commensals present in the SM communities, we did not detect an increase in the relative abundances of these bacteria, but rather saw an expansion of Bacteroides uniformis (Fig EV2a), which can be explained by its demonstrated capacity to subsist on this substrate (La Rosa, Kachrimanidou, et al., 2019).

• Figure 4d - what were the exact enzymes looked at here - put these in the figure
The enzyme names are listed directly above the plot. We have increased the font size for improved readability. The exact names according to the UniRef90 identifiers (obtained using the "rename" function in the biobakery3 HUMAnN pipeline) can be found in the accompanying Source Data file.

• In some of the data sets in the supplementary the layout means that the tables are all over the place - put these into excel files? It wasn't possible for me to look at the raw data myself
While we understand this request, we note that the journal guidelines ask us to split the tables into individual files. We expect that the inconvenience would be minimized in the online publication as the datasets are then hyperlinked where referenced.

• My only extra suggestion would be a table somewhere summarising the results: the different SMs vs the different parameters. See what you think.
We appreciate the suggestion and did indeed attempt to make such a table, however we found that the summary table was unable to capture some of the nuances of showing the data in the figures. Therefore, and in consultation with the editor, we agreed not to include such a summary table at this time.

3rd Apr 2024

Manuscript Number: MSB-2023-12163R

Title: Diet-driven differential response of Akkermansia muciniphila modulates pathogen susceptibility

Dear Mahesh,

Thank you for sending us your revised manuscript. We have evaluated your revision and we think that all issues raised by the reviewers have been addressed. I am glad to inform you that we can soon accept the manuscript for publication, pending some final editorial requests listed below.

- Our data editors have noted that the following needs to be edited in the figure legends:
 - Please define the annotated p values *****/**/*** in the legend of figure EV 2b, g; as appropriate.
 - Please indicate the statistical test used for data analysis in the legends of figures 3d-e; 5a, c-d; EV 5.
 - In figure EV 2d there is a mismatch between the annotated p values in the figure legend and the annotated p values in the figure file.
 - Please include information related to n in the legends of figures 2b-c; 3a; 4d; EV 2b-d; EV 3a.
 - Although 'n' is provided, please describe the nature of entity for 'n' (biological? Technical?) in the legend of figure 5b.
 - Please define the error bars in the legends of figures 2b-c; 5b.
 - The measure of center for the error bars needs to be defined in the legends of figures 1f; 2e-g; 3a; 4a-b, d.
- The funding information provided in the manuscript text (Acknowledgements) should match the information entered in the online submission system. Currently the FNR BRIDGES grant (22/17426243) is missing from the submission system.
- The References should be formatted according to the Molecular Systems Biology reference style (i.e., ordered alphabetically and listing the first 10 authors followed by et al).
- Please remove the 'Authors Contributions' from the manuscript. The 'Author Contributions' section is replaced by the CRediT contributor roles taxonomy to specify the contributions of each author in the journal submission system. Please use the free text box in the 'author information' section of the online submission system to provide more detailed descriptions if needed (e.g., 'X provided intracellular Ca⁺⁺ measurements in fig Y').
- Please provide the Appendix as a PDF.
- The Source Data files need to be reorganized. Please provide one Source Data file (xls or zip folder) per figure. All source data for Appendix and/or EV Figures should be included in a single zip folder. If some Source Data are the same for multiple figures, there is no need to provide them twice. If they have been provided as EV Datasets, there is no need to provide them again or to reference the EV Dataset.
- The Table legend should be renamed from Appendix Table 1 to Table EV1.

Please resubmit your revised manuscript online **within one month** and ideally as soon as possible. If we do not receive the revised manuscript within this time period, the file might be closed and any subsequent resubmission would be treated as a new manuscript. Please use the Manuscript Number (above) in all correspondence.

Click on the link below to submit your revised paper.

Kind regards,

Maria

Maria Polychronidou, PhD
Senior Editor
Molecular Systems Biology

If you do choose to resubmit, please click on the link below to submit the revision online before 3rd May 2024.

IMPORTANT: Please note that corresponding authors are required to supply an ORCID ID for their name upon submission of a revised manuscript (EMBO Press signed a joint statement to encourage ORCID adoption).
(<https://www.embopress.org/page/journal/17444292/authorguide#editorialprocess>)
Currently, our records indicate that the ORCID for your account is 0000-0002-9223-2209.

Please click the link below to modify this ORCID:
Link Not Available

*** PLEASE NOTE *** As part of the EMBO Press transparent editorial process initiative (see our Editorial at <https://dx.doi.org/10.1038/msb.2010.72> , Molecular Systems Biology will publish online a Review Process File to accompany accepted manuscripts. When preparing your letter of response, please be aware that in the event of acceptance, your cover letter/point-by-point document will be included as part of this File, which will be available to the scientific community. More information about this initiative is available in our Instructions to Authors. If you have any questions about this initiative, please contact the editorial office (msb@embo.org).

All editorial and formatting issues were resolved by the authors.

12th Apr 2024

Manuscript number: MSB-2023-12163RR

Title: Diet-driven differential response of Akkermansia muciniphila modulates pathogen susceptibility

Dear Mahesh,

Thank you again for sending us your revised manuscript. We are now satisfied with the modifications made and I am pleased to inform you that your paper has been accepted for publication.

Kind regards,

Maria

Maria Polychronidou, PhD
Senior Editor
Molecular Systems Biology
